# Out of time ordered effective dynamics of a quartic oscillator

**Bidisha Chakrabarty[⋆] and Soumyadeep Chaudhuri[†]**

International Centre for Theoretical Sciences (ICTS-TIFR),
Tata Institute of Fundamental Research, Shivakote, Hesaraghatta, Bangalore 560089, India

⋆ bidisha.chakrabarty@icts.res.in, † soumyadeep.chaudhuri@icts.res.in

## Abstract

We study the dynamics of a quantum Brownian particle weakly coupled to a thermal bath. Working in the Schwinger-Keldysh formalism, we develop an effective action of the particle up to quartic terms. We demonstrate that this quartic effective theory is dual to a stochastic dynamics governed by a non-linear Langevin equation. The Schwinger-Keldysh effective theory, or the equivalent non-linear Langevin dynamics, is insufficient to determine the out of time order correlators (OTOCs) of the particle. To overcome this limitation, we construct an extended effective action in a generalised Schwinger-Keldysh framework. We determine the additional quartic couplings in this OTO effective action and show their dependence on the bath's 4-point OTOCs. We analyse the constraints imposed on the OTO effective theory by microscopic reversibility and thermality of the bath. We show that these constraints lead to a generalised fluctuation-dissipation relation between the non-Gaussianity in the distribution of the thermal noise experienced by the particle and the thermal jitter in its damping coefficient. The quartic effective theory developed in this work provides extension of several results previously obtained for the cubic OTO dynamics of a Brownian particle.

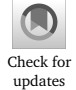

# 1 Introduction

The effective theory framework is a very useful tool in understanding the dynamics of open quantum systems. Such an open system typically interacts with a large environment which has a complicated dynamics. This complexity of the environment often makes a microscopic analysis practically impossible for studying the observables of the system. In such a scenario, one can try to develop an effective theory for only the infrared (low-frequency) degrees of freedom of the system.

The construction of such an effective theory usually relies on imposing

  a) few general conditions based on the unitarity of the microscopic theory, and

  b) some special conditions based on the symmetries in the particular theory.

After identifying these conditions, one can write down the most general effective theory that satisfies them. This theory can be then used to determine the observables of the system in terms of the effective couplings.

A paradigmatic example of an open system where such an effective dynamics can be worked out is a Brownian particle interacting with a thermal bath. The first step towards constructing the effective theory is to consider the evolution of the density matrix of the (particle+bath) combined system. One can then trace out the bath's degrees of freedom to get the evolution of the reduced density matrix of the particle.

An efficient way to obtain the evolution of the reduced density matrix is via a path integral formalism developed by Feynman-Vernon [1], Schwinger [2] and Keldysh [3], which is now commonly known as the 'Schwinger-Keldysh formalism' [4–6]. In this formalism, tracing out the bath's degrees of freedom amounts to integrating over them to obtain a correction to the particle's action. This correction, called the 'influence phase' [1], encapsulates the entire contribution of the interaction with the bath to the effective dynamics of the particle. As we will discuss later in section 3.2, the form of the influence phase is completely determined in terms of correlators of the bath operator that couples to the particle.

Generally, such thermal correlators of the bath are hard to determine. Nevertheless, the relationship between the bath's correlators and the particle's influence phase is interesting for the following reasons:

1. Such a relation provides a way to extract information about the bath's correlators by studying the effective dynamics of the particle.

2. The thermality of the bath leads to the Kubo-Martin-Schwinger (KMS) relations [7–10] between its correlators. These relations, in turn, constrain the effective dynamics of the particle.

3. There may be some symmetries in the bath's dynamics which would lead to further relations between its correlators. Such relations would impose additional constraints on the particle's effective theory.

4. It may be possible to construct simple toy models where one has analytic handle on the microscopic dynamics of the bath. In such cases, one can calculate the bath's correlators and then determine the effective couplings of the particle from them. Insights obtained from studying such simple toy models can be used to get a general idea about the dynamics of the particle in more complicated situations.

A classic example of such a simple toy model was provided by Caldeira and Leggett in [11] where they considered a bath comprising of a set of harmonic oscillators. These oscillators couple to the particle through an interaction which is bilinear in the positions of the particle and the bath oscillators. In this setup, Caldeira-Leggett integrated out the bath oscillators to obtain a temporally non-local influence phase which is quadratic in the particle's position. The simplicity of this quadratic effective theory makes it possible to employ both analytic and numerical techniques to study the evolution of the particle [11–21] . Such studies have shown that, even in this simple scenario, the Brownian particle demonstrates a rich variety of phenomena such as decoherence, dissipation, thermalisation, etc.

The Caldeira-Leggett model is also suitable for exploring a regime where the particle's dynamics is approximately local in time. Such a local dynamics can be obtained when the effect of the interaction with the particle dissipates in the bath much faster than the time-scale in which the particle evolves. As discussed in [22], this can be ensured by taking the temperature of the bath to be very high and choosing an appropriate distribution for the couplings between the particle and the bath oscillators. This regime in which the particle has an approximately local dynamics is often called the 'Markovian limit'.

In this limit, the quadratic effective theory of the Brownian particle is dual to a classical stochastic dynamics governed by a linear Langevin equation with a Gaussian noise [6]. Under this Langevin dynamics, the particle experiences a damping as well as a randomly fluctuating force. Both these forces originate from the interaction with the bath, and hence their strengths are related to each other. As we will review in section 4.4, this fluctuation-dissipation relation [23–26] follows from the KMS relations between the 2-point correlators of the bath.

Many of the features of the Caldeira-Leggett model that we discussed above are shared by more complicated systems where the quadratic effective theory holds to a good approximation [22, 27–29]. Nevertheless, it is interesting to explore the corrections to this effective dynamics by including the cubic and higher degree terms in the analysis. A simple way to obtain such corrections is to extend the Caldeira-Leggett model by introducing a non-linear combination of the bath oscillators' positions in the operator that couples to the particle [29, 30].

One such extension was discussed in [30] where the oscillators in the bath were divided into two sets (labelled by X and Y). Then apart from the bilinear interactions (which are present in the Caldeira-Leggett model), small cubic interactions were introduced between the particle

(q) and pairs of bath oscillators. Each such pair consisted of one oscillator drawn from the set X and the other from the set Y. Due to these cubic interactions, this model was called the 'qXY model'.

Introduction of these cubic interactions leads to cubic and higher degree terms in the effective action of the particle. In [30], this effective theory was studied up to the cubic terms in a Markovian limit. It was shown to be dual to a non-linear Langevin dynamics which has perturbative corrections over the linear Langevin dynamics in the Caldeira-Leggett model. The additional parameters in this non-linear dynamics receive contributions from the 3-point correlators of the bath. These 3-point correlators lead to an anharmonicity in the particle's dynamics and a non-Gaussianity in the distribution of the thermal noise experienced by the particle. In addition, they also give rise to thermal jitters[1] in both the frequency and the damping of the particle. The thermal jitter in the damping coefficient and the non-Gaussianity in the noise are connected by a generalised fluctuation-dissipation relation [30].

A valuable insight gained from the cubic effective theory in [30] is that this generalised fluctuation-dissipation relation arises from a combined effect of the thermality of the bath and the microscopic time-reversal invariance in its dynamics. These features of the bath manifest in the form of certain relations between its correlators [2]. As discussed in [9] and [30], these relations between thermal correlators lead to the inclusion of Out-of-Time-Order Correlators (OTOCs) [31–33] of the bath in the analysis. Such thermal OTOCs have aroused significant interest in the recent years as they provide insight into the chaotic behaviour of quantum systems [34–40], and serve as useful diagnostic measures in the study of thermalisation and many-body localisation [41–44].

A convenient way to study the effects of the bath's OTOCs on the particle's dynamics is to extend the effective theory framework to include computation of the particle's OTOCs. Such an OTO effective theory was developed upto cubic terms in [45]. There, it was seen that this extension requires introduction of some new couplings which are determined by the 3-point OTOCs of the bath. These new couplings in the cubic OTO effective theory get related to couplings in the Schwinger-Keldysh effective theory (or the equivalent non-linear Langevin dynamics) due to microscopic reversibility [3] in the bath [30]. At the same time, the KMS relations between bath's correlators connect the non-Gaussianity in the noise to one of the OTO couplings [30]. Combining these relations, one obtains the generalised fluctuation-dissipation relation mentioned above.

In this paper, we extend the analysis of the OTO effective theory of the Brownian particle by including the quartic terms in the effective action. There are three reasons for developing this extension. We enumerate them below.

1. Such an extension lays the ground for a convenient framework to compute 4-point OTO correlators in open quantum systems. This may lead to further insight into the role played by these OTOCs in physical phenomena.

2. The quartic couplings in this extended OTO effective theory receives contributions from the 4-point thermal OTOCs of the bath. Such thermal OTOCs have been the focus of most recent studies on chaos in quantum systems [34–40]. Including the contributions of these 4-point OTOCs in the effective theory of the particle opens the possibility of probing the chaotic behaviour of the bath via the quartic effective couplings [45] (see the discussion in section 5).

---

[1] These thermal jitters arise from the same non-Gaussian noise distribution.

[2] For example, the thermality of the bath leads to the KMS relations that we mentioned earlier.

[3] Such relations between cubic effective couplings can be interpreted as generalisations of the Onsager-Casimir reciprocal relations [46–48] in quadratic effective theories for systems with multiple degrees of freedom.

3. The relations between the quartic effective couplings and the 4-point OTOCs of the bath allows one to extend the analysis of the constraints imposed on the effective dynamics of the particle due to the bath's thermality and microscopic reversibility.

To present the construction of the effective theory with a concrete example, we choose to work with the qXY model that we discussed earlier. We simplify this model a bit by switching off the Caldeira-Legget-like bilinear interactions. This leads to a symmetry in the particle's dynamics under

$$q \to -q, \tag{1}$$

which results in the vanishing of all the odd degree terms in the effective action.

For this simplified qXY model, we first develop the Schwinger-Keldysh effective theory of the particle up to quartic terms. Working in a Markovian limit, we determine the dependence of the quartic couplings on the 4-point Schwinger-Keldysh correlators of the bath. Then we show that this quartic effective theory is dual to a non-linear Langevin dynamics with a non-Gaussian noise distribution. This stochastic dynamics has a structure quite similar to the one discussed in [30] for the cubic effective theory.

We then extend the analysis to the OTO dynamics of the particle by determining its effective action on a contour with two time-folds[4]. The form of this effective action is constrained by the microscopic unitarity in the dynamics of the (particle+bath) combined system [45]. We figure out all the additional OTO couplings consistent with these constraints, and then determine their dependence on the 4-point OTO correlators of the bath. As in case of the 3-point correlators [30] , these 4-point functions of the bath satisfy certain relations imposed by microscopic time-reversal invariance and thermality. We show that these relations between the bath's correlators lead to some constraints on the particle's effective couplings. These constraints can be interpreted as OTO generalisations [30] of the Onsager reciprocal relations [46, 47] and the fluctuation-dissipation relation [25, 49]. Combining these constraints, one can obtain a generalised fluctuation-dissipation relation between two quartic couplings in the Schwinger-Keldysh effective theory (or equivalently, the dual non-linear Langevin dynamics). Just as in case of the cubic effective theory [30], we find that this generalised fluctuation-dissipation relation connects the thermal jitter in the damping coefficient of the particle and the non-Gaussianity in the noise distribution.

**Organisation of the paper**

The structure of this paper is as follows:

In section 2, we briefly describe the qXY model which serves as a concrete example in our analysis.

In section 3, we develop the Schwinger-Keldysh effective action for the particle in a Markovian regime. We determine the relations between the effective couplings and the Schwinger-Keldysh correlators of the bath. We also demonstrate a duality between the Schwinger-Keldysh effective theory and a stochastic dynamics governed by a non-linear Langevin equation.

In section 4, we extend the effective theory framework to include OTOCs. We determine all the additional quartic OTO couplings appearing in this extension. Exploring the constraints imposed on the OTO effective theory by the thermality of the bath and its microscopic reversibility, we derive the generalised Onsager relations as well as a generalised fluctuation-dissipation relation between the quartic effective couplings. By computing the effective couplings for the qXY model, we verify that all these relations are indeed satisfied.

In section 5, we conclude with some discussion on future directions.

---

[4]Such a path integral formalism on a contour with multiple time-folds is a straightforward generalisation [32, 33, 45] of the Schwinger-Keldysh formalism.

In appendix A, we show how the cumulants of the bath's correlators decay when the interval between any two insertions is increased. In appendix B, we provide an argument for the validity of Markov approximation in a certain parameter regime. In appendix C, we express the quartic OTO couplings in terms of the 4-point OTO cumulants of the bath. In appendix D, we provide the forms of the high-temperature limit of all the quartic couplings in terms of the thermal spectral functions of the bath.

## 2 Description of the $qXY$ model

In this section, we describe the $qXY$ model which will serve as a concrete example for developing the particle's effective dynamics in the rest of the paper.

In this model, a Brownian particle interacts with a thermal bath (at the temperature $\frac{1}{\beta}$) [5] comprising of two sets of harmonic oscillators. We represent the positions of the oscillators in these two sets by $X^{(i)}$ and $Y^{(j)}$, and the position of the Brownian particle by $q$. The particle and the bath couple via cubic interactions involving two of the bath oscillators, one taken from each set. The Lagrangian of this model is given by

$$
\begin{aligned}
L[q, X, Y] = {} & \frac{m_{p0}}{2}(\dot{q}^2 - \bar{\mu}_0^2 q^2) + \sum_i \frac{m_{x,i}}{2}(\dot{X}^{(i)2} - \mu_{x,i}^2 X^{(i)2}) \\
& + \sum_j \frac{m_{y,j}}{2}(\dot{Y}^{(j)2} - \mu_{y,j}^2 Y^{(j)2}) + \lambda \sum_{i,j} g_{xy,ij} X^{(i)} Y^{(j)} q.
\end{aligned}
\tag{2}
$$

The bath operator that couples to the particle is

$$
\lambda O \equiv \lambda \sum_{i,j} g_{xy,ij} X^{(i)} Y^{(j)}.
\tag{3}
$$

Notice that all odd point correlators of this operator vanish in the thermal state. We will see later that this leads to vanishing of all odd degree terms in the effective action of the particle. Among the remaining terms, we would like to restrict our attention to only the quadratic and the quartic ones in this paper. We will see that the couplings corresponding to these terms receive contributions from the connected parts of the 2-point and 4-point correlators of $O$ at leading order in $\lambda$. So, we need to compute these correlators to obtain the leading order forms of the quadratic and the quartic couplings of the particle.

While computing the correlators, we assume that there is a large number of oscillators in the bath, and the frequencies of these oscillators are densely distributed. In such a situation, one can go to the continuum limit of this distribution, and replace the sum over the frequencies by integrals in the following way:

$$
\sum_{i,j} \frac{g_{xy,ij}^2}{m_{x,i} m_{y,j}} \to \int_0^\infty \frac{d\mu_x}{2\pi} \int_0^\infty \frac{d\mu_y}{2\pi} \left\langle\!\!\left\langle \frac{g_{xy}^2(\mu_x, \mu_y)}{m_x m_y} \right\rangle\!\!\right\rangle,
\tag{4}
$$

$$
\begin{aligned}
& \sum_{i_1, j_1} \sum_{i_2, j_2} \frac{g_{xy,i_1 j_1} g_{xy,i_1 j_2} g_{xy,i_2 j_1} g_{xy,i_2 j_2}}{m_{x,i_1} m_{y,j_1} m_{x,i_2} m_{y,j_2}} \\
& \to \int_0^\infty \frac{d\mu_x}{2\pi} \int_0^\infty \frac{d\mu_y}{2\pi} \int_0^\infty \frac{d\mu_x'}{2\pi} \int_0^\infty \frac{d\mu_y'}{2\pi} \left\langle\!\!\left\langle \frac{g_{xy}(\mu_x, \mu_y) g_{xy}(\mu_x, \mu_y') g_{xy}(\mu_x', \mu_y) g_{xy}(\mu_x', \mu_y')}{m_x m_y m_x' m_y'} \right\rangle\!\!\right\rangle,
\end{aligned}
\tag{5}
$$

---

[5] Here we are working in units where the Boltzmann constant $k_B = 1$.

where

$$\left\langle\!\!\left\langle \frac{g_{xy}^2(\mu_x,\mu_y)}{m_x m_y} \right\rangle\!\!\right\rangle \equiv \sum_{i,j} \frac{g_{xy,ij}^2}{m_{x,i} m_{y,j}} \Big[ 2\pi\delta(\mu_x - \mu_{x,i}) 2\pi\delta(\mu_y - \mu_{y,j}) \Big], \tag{6}$$

$$\left\langle\!\!\left\langle \frac{g_{xy}(\mu_x,\mu_y) g_{xy}(\mu_x,\mu_y') g_{xy}(\mu_x',\mu_y) g_{xy}(\mu_x',\mu_y')}{m_x m_y m_x' m_y'} \right\rangle\!\!\right\rangle$$

$$\equiv \sum_{i_1,j_1} \sum_{i_2,j_2} \frac{g_{xy,i_1 j_1} g_{xy,i_1 j_2} g_{xy,i_2 j_1} g_{xy,i_2 j_2}}{m_{x,i_1} m_{y,j_1} m_{x,i_2} m_{y,j_2}} \Big[ 2\pi\delta(\mu_x - \mu_{x,i_1}) 2\pi\delta(\mu_y - \mu_{y,j_1}) \Big] \tag{7}$$

$$\Big[ 2\pi\delta(\mu_x' - \mu_{x,i_2}) 2\pi\delta(\mu_y' - \mu_{y,j_2}) \Big].$$

We choose the functions $\left\langle\!\!\left\langle \frac{g_{xy}^2(\mu_x,\mu_y)}{m_x m_y} \right\rangle\!\!\right\rangle$ and $\left\langle\!\!\left\langle \frac{g_{xy}(\mu_x,\mu_y) g_{xy}(\mu_x,\mu_y') g_{xy}(\mu_x',\mu_y) g_{xy}(\mu_x',\mu_y')}{m_x m_y m_x' m_y'} \right\rangle\!\!\right\rangle$ in a manner which would give us an approximately local effective dynamics of the particle. As we will see, such a local dynamics can be obtained if the time-scales involved in the evolution of the particle are much larger than the time-scale in which cumulants of the operator $O(t)$ decay. Keeping this in mind, we choose the distribution of the couplings to satisfy

$$\lambda^2 \left\langle\!\!\left\langle \frac{g_{xy}^2(\mu_x,\mu_y)}{m_x m_y} \right\rangle\!\!\right\rangle = \Gamma_2 \frac{4\mu_x^2 \Omega^2}{\mu_x^2 + \Omega^2} \frac{4\mu_y^2 \Omega^2}{\mu_y^2 + \Omega^2}, \tag{8}$$

$$\lambda^4 \left\langle\!\!\left\langle \frac{g_{xy}(\mu_x,\mu_y) g_{xy}(\mu_x,\mu_y') g_{xy}(\mu_x',\mu_y) g_{xy}(\mu_x',\mu_y')}{m_x m_y m_x' m_y'} \right\rangle\!\!\right\rangle$$

$$= \Gamma_4 \Big( \frac{4\mu_x^2 \Omega^2}{\mu_x^2 + \Omega^2} \Big) \Big( \frac{4\mu_y^2 \Omega^2}{\mu_y^2 + \Omega^2} \Big) \Big( \frac{4\mu_x'^2 \Omega^2}{\mu_x'^2 + \Omega^2} \Big) \Big( \frac{4\mu_y'^2 \Omega^2}{\mu_y'^2 + \Omega^2} \Big), \tag{9}$$

where $\Omega$ is a UV regulator.

For this distribution of the couplings, we study the high temperature limit of the 2-point and 4-point cumulants of $O(t)$ in appendix A. There, we show that when the time intervals between the insertions are increased, these cumulants decay exponentially at rates which are of the order of $\Omega$. If the natural frequency $(\overline{\mu}_0)$ of the particle and the frequency scales associated with the parameters $\Gamma_2$ and $\Gamma_4$ are taken to be much smaller than $\Omega$, then the bath correlators decay much faster than the rate at which the particle evolves. This ensures that the effect (via the bath) of some earlier state of the particle on its later dynamics is heavily suppressed. Consequently, we get an approximately local effective dynamics of the particle. We discuss this effective dynamics in the next section.

# 3 Schwinger-Keldysh effective theory of the oscillator

In this section, we develop the Schwinger-Keldysh effective theory for the dynamics of the particle. We also demonstrate a duality between this quantum effective theory and a classical stochastic theory governed by a non-linear Langevin equation.

## 3.1 Evolution of the reduced density matrix of the particle

Let us begin the discussion on the particle's effective dynamics by specifying the initial state of the particle and reviewing a path integral formalism which gives its evolution.

Consider the situation where the particle is initially unentangled with the bath. Suppose the cubic interaction given in (2) is switched on at a time $t_0$. Then the state of the (particle+bath) combined system at $t_0$ is given by[6]

$$\rho_0 = \frac{e^{-\beta H_B}}{Z_B} \otimes \rho_{p0}, \tag{10}$$

where $H_B$ and $Z_B$ are the Hamiltonian and the partition function of the bath respectively, and $\rho_{p0}$ is the density matrix of the particle at the time $t_0$.

After the particle starts interacting with the bath, an effective description of its state can be given in terms of its reduced density matrix which is obtained by tracing out the bath's degrees of freedom in the density matrix of the combined system. The evolution of the reduced density matrix is given by the quantum master equation [22] of the particle. An equivalent description of this evolution can be developed in terms of an effective action of the particle [7] in the Schwinger-Keldysh formalism [1–6].

In this formalism, one can first determine the density matrix of the combined system at some later time $t_f$ from a path integral on the contour shown in figure 1. This contour has two legs which we label as 1 and 2. For each of these legs, we need to take a copy of the degrees of

time

$t_0$        2        $t_f$

Figure 1: Contour for evolution of the density matrix

freedom of both the particle and the bath: $\{q_1, X_1, Y_1\}$ and $\{q_2, X_2, Y_2\}$.[8] The evolution of the density matrix of the combined system is then obtained from a path integral with a Lagrangian of the following form[9]

$$L_{SK} = L[q_1, X_1, Y_1] - L[-q_2, X_2, Y_2]. \tag{11}$$

Let us denote all the degrees of freedom of the combined system collectively by $Q$. Then, the two copies of these degrees of freedom can be expressed as

$$\{q_1, X_1, Y_1\} \to Q_1, \quad \{-q_2, X_2, Y_2\} \to Q_2. \tag{12}$$

Now, given the information that the initial density matrix at time $t_0$ is $\rho_0(Q_{10}, Q_{20})$, the density matrix at the later time $t_f$ is given by the following path integral:

$$\rho_f(Q_{1f}, Q_{2f}) = \int dQ_{10} \int dQ_{20}\, \rho_0(Q_{10}, Q_{20}) \int_{\substack{Q_1(t_0)=Q_{10}, \\ Q_2(t_0)=Q_{20}}}^{\substack{Q_1(t_f)=Q_{1f}, \\ Q_2(t_f)=Q_{2f}}} [DQ_1][DQ_2] e^{i \int_{t_0}^{t_f} dt\, L_{SK}[Q_1(t), Q_2(t)]}. \tag{13}$$

To get the reduced density matrix ($\rho_{pf}$) of the particle at the time $t_f$, we need to trace out the bath's degrees of freedom at $t_f$. This can be achieved in the above path integral by setting

$$X_{1f} = X_{2f} = X_f \text{ and } Y_{1f} = Y_{2f} = Y_f, \tag{14}$$

---

[6]Here, we work in units where $\hbar = 1$.

[7]We refer the reader to [50] for a detailed discussion on how the quantum master equation is related to the Schwinger-Keldysh effective action.

[8]Here, $X$ and $Y$ denote collectively all the oscillators in the two sets.

[9]Here we find it convenient to put an extra minus sign in $q_2$ over the standard convention followed in texts on the Schwinger-Keldysh formalism [4, 5]. This is consistent with the convention followed in [30, 45].

and then integrating over $X_f$ and $Y_f$.

This path integral representation of the reduced density matrix still involves integrals over the bath's degrees of freedom. One can, however, express this integral in terms of only the particle's degrees of the freedom by integrating out the bath's coordinates. As mentioned in the introduction, such an intergral over the bath's coordinates leads to a correction to the particle's action. This additional piece in the action which encapsulates the influence of the bath on the particle's dynamics is called the 'influence phase' of the particle [1]. Next, we discuss the form of this influence phase.

## 3.2 The influence phase of the particle

In the previous subsection we argued that the expression for the reduced density matrix $\rho_{pf}$ can be obtained from the path integral in (13) by identifying the bath's degrees of freedom on the two legs at the time $t_f$. Therefore, in this expression, the bath's coordinates have to be integrated over a contour of the following form:

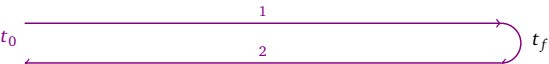

Figure 2: Schwinger-Keldysh contour

This is the usual Schwinger-Keldysh contour where the two legs are connected by a future-turning point. Path integrals over such a contour give correlators where the insertions are contour-ordered according to the arrow indicated in figure 2. Therefore, such contour-ordered correlators of the bath would contribute to the effective action of the particle when the bath's coordinates are integrated out.

The contributions of the bath correlators are imprinted in the influence phase $(W_{SK})$ which appears in the path integral for $\rho_{pf}$ as follows

$$\rho_{pf}(q_{1f}, -q_{2f}) = \int dq_{10} \int dq_{20}\, \rho_{p0}(q_{10}, -q_{20})$$

$$\int_{\substack{q_1(t_0)=q_{10}, \\ q_2(t_0)=q_{20}}}^{\substack{q_1(t_f)=q_{1f}, \\ q_2(t_f)=q_{2f}}} [Dq_1][Dq_2] e^{i\left[\frac{1}{2}m_{p0}\int_{t_0}^{t_f} dt\, \left\{(\dot{q}_1^2 - \bar{\mu}_0^2 q_1^2) - (\dot{q}_2^2 - \bar{\mu}_0^2 q_2^2)\right\} + W_{SK}\right]}. \tag{15}$$

The influence phase in the above expression can be expanded as a perturbation series in $\lambda$:

$$W_{SK} = \sum_{n=1}^{\infty} \lambda^n W_{SK}^{(n)}, \tag{16}$$

where

$$W_{SK}^{(n)} = i^{n-1} \sum_{i_1, \cdots, i_n=1}^{2} \int_{t_0}^{t_f} dt_1 \cdots \int_{t_0}^{t_{n-1}} dt_n \, \langle \mathcal{T}_C O_{i_1}(t_1) \cdots O_{i_n}(t_n) \rangle_c \, q_{i_1}(t_1) \cdots q_{i_n}(t_n). \tag{17}$$

Here $\langle \mathcal{T}_C O_{i_1}(t_1) \cdots O_{i_n}(t_n) \rangle_c$ is the cumulant (connected part) of a contour-ordered correlator of $O(t)$ where the insertion at time $t_j$ is on the $i_j^{\text{th}}$ leg.

We remind the reader that, for the $qXY$ model, such a thermal correlator of $O(t)$ is zero when the number of insertions is odd. This in turn means that all the odd degree terms in

the perturbative expansion of the influence phase vanish. Among the remaining terms, we restrict our attention to those whose coefficients are up to $O(\lambda^4)$. This leaves us with only the quadratic and quartic terms whose expressions are given below:

$$W_{SK}^{(2)} = i \sum_{i_1,i_2=1}^{2} \int_{t_0}^{t_f} dt_1 \int_{t_0}^{t_1} dt_2 \, \langle \mathcal{T}_C O_{i_1}(t_1) O_{i_2}(t_2) \rangle_c \, q_{i_1}(t_1) q_{i_2}(t_2), \quad (18)$$

$$W_{SK}^{(4)} = -i \sum_{i_1,\cdots,i_4=1}^{2} \int_{t_0}^{t_f} dt_1 \int_{t_0}^{t_1} dt_2 \int_{t_0}^{t_2} dt_3 \int_{t_0}^{t_3} dt_4 \, \langle \mathcal{T}_C O_{i_1}(t_1) O_{i_2}(t_2) O_{i_3}(t_3) O_{i_4}(t_4) \rangle_c$$
$$q_{i_1}(t_1) q_{i_2}(t_2) q_{i_3}(t_3) q_{i_4}(t_4) \,. \quad (19)$$

Plugging these expressions for the quadratic and quartic terms in the influence phase into the path integral given in (15), one can determine the evolution of the reduced density matrix of the particle. The effective action that appears in this path integral can also be used to compute the correlators of the particle.[10] From the form of the influence phase in (17), we can see that this effective action is non-local in time. Next, we discuss a certain limit in which one may get an approximately local form for this effective action.

## 3.3 Markovian limit

In appendix A, we have shown that, in the high temperature limit ($\beta\Omega \ll 1$), the cumulants of the operator $O(t)$ appearing in (18) and (19) decay exponentially when the separation between any two insertions is increased. We have also shown that the decay rates of these cumulants are of the order of the cut-off frequency $\Omega$. Then the values of the coefficient functions multiplying the q's at different times in (18) and (19) become negligible when the interval between any two instants is $O(\Omega^{-1})$.

We choose to work in a regime where this time-scale ($\Omega^{-1}$) is much smaller than all the time-scales involved in the evolution of the particle. To ensure this, we take the parameters in the qXY model to satisfy the following conditions:[11]

$$\beta\Omega \ll 1, \ \overline{\mu}_0 \ll \Omega, \ \Gamma_2 \ll \beta(\beta\Omega), \ \Gamma_4 \ll (\Gamma_2)^2 \,. \quad (20)$$

In this regime, the bath's cumulants die out too fast to transmit any significant effect of the history of the particle on its dynamics at a later instant. Consequently, one can get an approximately local form for the influence phase by Taylor-expanding the q's at different instants around $t_1$:

$$W_{SK}^{(2)} \approx i \sum_{i_1,i_2=1}^{2} \int_{t_0}^{t_f} dt_1 \Big[ \Big\{ \int_{t_0}^{t_1} dt_2 \, \langle \mathcal{T}_C O_{i_1}(t_1) O_{i_2}(t_2) \rangle_c \Big\} q_{i_1}(t_1) q_{i_2}(t_1 - \epsilon)$$
$$+ \Big\{ \int_{t_0}^{t_1} dt_2 \, \langle \mathcal{T}_C O_{i_1}(t_1) O_{i_2}(t_2) \rangle_c \, t_{21} \Big\} q_{i_1}(t_1) \dot{q}_{i_2}(t_1 - \epsilon) \quad (21)$$
$$+ \Big\{ \int_{t_0}^{t_1} dt_2 \, \langle \mathcal{T}_C O_{i_1}(t_1) O_{i_2}(t_2) \rangle_c \, \frac{t_{21}^2}{2} \Big\} q_{i_1}(t_1) \ddot{q}_{i_2}(t_1 - \epsilon) \Big],$$

---

[10]To compute the particle's correlators from path integrals with this effective action, one needs to put $q_1 = -q_2$ at some time in the future of all the insertions.

[11]In appendix B, we provide an argument for the validity of Markov approximation in this regime.

$$W_{SK}^{(4)} \approx -i \sum_{i_1,\cdots,i_4=1}^{2} \int_{t_0}^{t_f} dt_1 \Big[ \Big\{ \int_{t_0}^{t_1} dt_2 \int_{t_0}^{t_2} dt_3 \int_{t_0}^{t_3} dt_4 \, \langle \mathcal{T}_C O_{i_1}(t_1) O_{i_2}(t_2) O_{i_3}(t_3) O_{i_4}(t_4) \rangle_c \Big\}$$

$$q_{i_1}(t_1) q_{i_2}(t_1-\epsilon) q_{i_3}(t_1-2\epsilon) q_{i_4}(t_1-3\epsilon)$$

$$+ \Big\{ \int_{t_0}^{t_1} dt_2 \int_{t_0}^{t_2} dt_3 \int_{t_0}^{t_3} dt_4 \, \langle \mathcal{T}_C O_{i_1}(t_1) O_{i_2}(t_2) O_{i_3}(t_3) O_{i_4}(t_4) \rangle_c \, t_{21} \Big\}$$

$$q_{i_1}(t_1) \dot{q}_{i_2}(t_1-\epsilon) q_{i_3}(t_1-2\epsilon) q_{i_4}(t_1-3\epsilon)$$

$$+ \Big\{ \int_{t_0}^{t_1} dt_2 \int_{t_0}^{t_2} dt_3 \int_{t_0}^{t_3} dt_4 \, \langle \mathcal{T}_C O_{i_1}(t_1) O_{i_2}(t_2) O_{i_3}(t_3) O_{i_4}(t_4) \rangle_c \, t_{31} \Big\}$$

$$q_{i_1}(t_1) q_{i_2}(t_1-\epsilon) \dot{q}_{i_3}(t_1-2\epsilon) q_{i_4}(t_1-3\epsilon)$$

$$+ \Big\{ \int_{t_0}^{t_1} dt_2 \int_{t_0}^{t_2} dt_3 \int_{t_0}^{t_3} dt_4 \, \langle \mathcal{T}_C O_{i_1}(t_1) O_{i_2}(t_2) O_{i_3}(t_3) O_{i_4}(t_4) \rangle_c \, t_{41} \Big\}$$

$$q_{i_1}(t_1) q_{i_2}(t_1-\epsilon) q_{i_3}(t_1-2\epsilon) \dot{q}_{i_4}(t_1-3\epsilon) \Big] . \tag{22}$$

Here, we have kept up to second order derivative terms in the quadratic piece to take into account the correction to the kinetic term in the particle's action. For the quartic piece, we have kept only terms with at most a single derivative. To preserve the information about the original ordering of the different time instants, we have kept a small point-split regulator $\epsilon > 0$.[12]

Using this approximate local form of the influence phase, one can determine the particle's correlators. The same correlators can be obtained from a 1-particle irreducible effective action [13] which is local in time. Next, we discuss the form of this 1-PI effective action.

## 3.4 The Schwinger-Keldysh 1-PI effective action

In the previous subsection we discussed an approximately local form of the influence phase which is obtained by perturbatively expanding the effects of the particle-bath interaction on the particle's dynamics. By putting appropriate initial conditions on path integrals with this influence phase one can calculate the correlators of the particle. However, as we saw, the influence phase has a slight non-locality due to the presence of the point-split regulator. It is cumbersome to keep track of this ordering of the time instants for the different q's in the effective action. To avoid this, we extend the framework introduced in [45] and develop a local 1-PI effective action for the particle which can be employed to compute its correlators.[14]

The form of this 1-PI effective action is constrained by some general principles which we discuss below.

- **Collapse rule:**

  The effective action should vanish under the following identification:

  $$q_1 = -q_2 = \tilde{q}. \tag{23}$$

---

[12]It is important to keep this regulator as otherwise one can get wrong answers while computing contributions of loop diagrams where two of the q's on the same vertex contract with each other.

[13]Only the tree level diagrams obtained from the 1-particle irreducible effective action contribute to the correlators of the particle.

[14]The issue of contraction of q's on the same vertex does not arise in the 1-PI effective theory framework because the correlators receive contributions only from tree-level diagrams. Hence, there is no need for a point-split regulator in the terms of the 1-PI effective action.

As discussed in [5, 45, 51], this condition is based on the fact that the particle is a part of a closed system governed by a unitary dynamics. At the level of correlators, it makes sure that the value of a contour-ordered correlator of the particle just picks up a sign when one slides the future-most insertion from one leg to the other without changing its temporal position. The sign is introduced because we are putting an extra minus sign in the definition of $q_2$ over the convention followed in [5, 51].

- **Reality conditions:**
  The effective action should become its own negative under complex conjugation of the effective couplings along with the following exchange:

$$q_1 \leftrightarrow -q_2. \tag{24}$$

  This condition is based on the Hermiticity of the operator q(t) which implies that the correlators of q should remain unchanged under a reversal of the ordering of the insertions followed by a complex conjugation. As discussed in [45, 51], the above reality conditions ensure that such relations between the particle's correlators are satisfied.

- **Symmetry under** $q_1 \to -q_1,\ q_2 \to -q_2$**:**
  The Lagrangian of the qXY mode given in (2) is symmetric under the following two transformations:.

$$
\begin{aligned}
q \to -q,\ X \to -X,\ Y \to Y,\\
q \to -q,\ X \to X,\ Y \to -Y.
\end{aligned}
\tag{25}
$$

  Moreover, the bath is in a thermal state which is also invariant under the above transformations. Now, if the initial state of the particle obeys the same symmetry, then all correlators of the particle with odd number of insertions would vanish. This vanishing of the odd point functions of the particle for a class of initial conditions can be ensured by demanding that the 1-PI effective action is symmetric under the following transformation:

$$q_1 \to -q_1\ ,\ q_2 \to -q_2\ . \tag{26}$$

The most general 1-PI effective Lagrangian that is consistent with these conditions can be written as a sum of terms with even number of q's as follows:

$$L_{\text{SK,1-PI}} = L_{\text{SK,1-PI}}^{(2)} + L_{\text{SK,1-PI}}^{(4)} + \cdots . \tag{27}$$

In this paper, we will focus only on the quadratic and the quartic terms in this effective action. We give the forms of these terms below:

**Quadratic terms:**

$$
\begin{aligned}
L_{\text{SK,1-PI}}^{(2)} =&\frac{1}{2}(\dot{q}_1^2 - \dot{q}_2^2) - \frac{i}{2}\, Z_I(\dot{q}_1 + \dot{q}_2)^2 - \frac{1}{2}\bar{\mu}^2(q_1^2 - q_2^2)\\
&+ \frac{i}{2}\,\langle f^2\rangle(q_1 + q_2)^2 - \frac{1}{2}\gamma(q_1 + q_2)(\dot{q}_1 - \dot{q}_2)\ .
\end{aligned}
\tag{28}
$$

Here we consider all quadratic terms up to two derivatives acting on the q's.[15] We have included the double derivative terms to take into account the renormalisation of the kinetic term in the action. Such a renormalisation introduces a correction to the effective mass of the particle on top of the bare mass $m_{p0}$. After taking into account this correction, we choose to work in units where the renormalised mass of the particle is unity.

---

[15]We are using the convention introduced in [30] for the quadratic couplings.

**Quartic terms:**

$$
\begin{aligned}
L_{\text{SK,1-PI}}^{(4)} =& \frac{i\zeta_N^{(4)}}{4!}(q_1+q_2)^4 + \frac{\zeta_\mu^{(2)}}{2}(q_1+q_2)^2(q_1^2-q_2^2) - \frac{i\overline{\zeta}_3}{8}(q_1^2-q_2^2)^2 \\
&- \frac{\overline{\lambda}_4}{48}(q_1^2-q_2^2)(q_1-q_2)^2 + \frac{\zeta_\gamma^{(2)}}{2}(q_1+q_2)^3(\dot{q}_1-\dot{q}_2) \\
&- \frac{\overline{\lambda}_{4\gamma}}{48}(\dot{q}_1+\dot{q}_2)(q_1-q_2)^3 + \frac{i\overline{\zeta}_{3\gamma}}{4}(q_1+q_2)^2(q_1-q_2)(\dot{q}_1-\dot{q}_2)\,.
\end{aligned}
\tag{29}
$$

Among the quartic terms, we keep those with at most a single derivative acting on the q's.[16] This is consistent with the order at which we truncated the Taylor series expansion of the terms in the influence phase. The reality conditions imply that all the couplings introduced in (28) and (29) are real.

Now, using this form of the 1-PI effective action, one can calculate the particle's correlators and match them with the same correlators obtained from the approximately local form of the influence phase. The computation of the correlators from both the influence phase and the 1-PI effective action would require specifying the corresponding initial conditions of the particle at some time after the local dynamics has set in. We assume that these initial conditions are the same in the two approaches up to perturbative corrections in $\lambda$. This allows us to compare the leading order forms of the connected parts of 2-point and 4-point correlators of q obtained from the two approaches. Such a comparison would yield relations between the effective couplings and the cumulants of the bath's correlators up to leading order in $\lambda$. We enumerate these relations in (32), (33) and table 1 below.

While expressing these relations, we adopt the following notational conventions:

1. We denote the time interval between two instants $t_i$ and $t_j$ by

$$
t_{ij} \equiv t_i - t_j\,.
\tag{30}
$$

2. For connected parts of correlators with nested commutators/ anti-commutators , we put all the insertions within square brackets enclosed by angular brackets. The innermost operator in the nested structure is positioned left-most in the expression within the brackets. The operators that one encounters as one moves outwards through the nested structure are placed progressively rightwards in the expression. The position of each anti-commutator is indicated by a + sign. For example,

$$
\begin{aligned}
&\langle[12]\rangle \equiv \langle[O(t_1),O(t_2)]\rangle_c,\ \langle[12_+]\rangle \equiv \langle\{O(t_1),O(t_2)\}\rangle_c\,, \\
&\langle[1234]\rangle \equiv \langle[[[O(t_1),O(t_2)],O(t_3)],O(t_4)]\rangle_c\,, \\
&\langle[12_+34]\rangle \equiv \langle[[\{O(t_1),O(t_2)\},O(t_3)],O(t_4)]\rangle_c\,, \\
&\langle[12_+3_+4]\rangle \equiv \langle[\{\{O(t_1),O(t_2)\},O(t_3)\},O(t_4)]\rangle_c\,, \\
&\langle[12_+3_+4_+]\rangle \equiv \langle\{\{\{O(t_1),O(t_2)\},O(t_3)\},O(t_4)\}\rangle_c\,,\text{etc.}
\end{aligned}
\tag{31}
$$

---

[16]The terms without any derivative were identified for a scalar field theory in [51].

**Quadratic couplings at leading order in** $\lambda$**:** The dependence of all the quadratic couplings on the bath's correlators were obtained in [45]. We provide these expressions below:

$$Z_I = \frac{\lambda^2}{2} \lim_{t_1-t_0 \to \infty} \Big[ \int_{t_0}^{t_1} dt_2 \langle [12_+] \rangle t_{12}^2 \Big] + O(\lambda^4),$$

$$\langle f^2 \rangle = \lambda^2 \lim_{t_1-t_0 \to \infty} \Big[ \int_{t_0}^{t_1} dt_2 \langle [12_+] \rangle \Big] + O(\lambda^4),$$

$$\Delta \overline{\mu}^2 \equiv \overline{\mu}^2 - \overline{\mu}_0^2 = -i \, \lambda^2 \lim_{t_1-t_0 \to \infty} \Big[ \int_{t_0}^{t_1} dt_2 \langle [12] \rangle \Big] + O(\lambda^4),$$

$$\gamma = i \, \lambda^2 \lim_{t_1-t_0 \to \infty} \Big[ \int_{t_0}^{t_1} dt_2 \langle [12] \rangle t_{12} \Big] + O(\lambda^4).$$

(32)

**Quartic couplings at leading order in** $\lambda$**:** The leading order form of any quartic coupling $g$ can be expressed as follows:

$$g = \lambda^4 \lim_{t_1-t_0 \to \infty} \int_{t_0}^{t_1} dt_2 \int_{t_0}^{t_2} dt_3 \int_{t_0}^{t_3} dt_4 \, \mathcal{I}[g] + O(\lambda^6) .$$

(33)

We provide the forms of the integrand $\mathcal{I}[g]$ for the quartic couplings in table 1.

Table 1: Relations between the SK 1-PI effective couplings and the correlators of $O(t)$

| $g$ | $\mathcal{I}[g]$ |
|---|---|
| $\zeta_N^{(4)}$ | $-3\langle [12_+3_+4_+] \rangle$ |
| $\overline{\lambda}_4$ | $6i\langle [1234] \rangle$ |
| $\overline{\lambda}_{4\gamma}$ | $2i(t_{12} + t_{13} + t_{14})\langle [1234] \rangle$ |
| $\zeta_\mu^{(2)}$ | $-\frac{i}{4}\Big( \langle [123_+4_+] \rangle + \langle [12_+34_+] \rangle + \langle [12_+3_+4] \rangle \Big)$ |
| $\zeta_\gamma^{(2)}$ | $\frac{i}{12}\Big( (3t_{12} - t_{13} - t_{14})\langle [123_+4_+] \rangle + (-t_{12} + 3t_{13} - t_{14})\langle [12_+34_+] \rangle + (-t_{12} - t_{13} + 3t_{14})\langle [12_+3_+4] \rangle \Big)$ |
| $\overline{\zeta}_3$ | $\Big( \langle [12_+34] \rangle + \langle [123_+4] \rangle + \langle [1234_+] \rangle \Big)$ |
| $\overline{\zeta}_{3\gamma}$ | $\frac{1}{2}\Big( (-t_{12} + t_{13} + t_{14})\langle [12_+34] \rangle + (t_{12} - t_{13} + t_{14})\langle [123_+4] \rangle + (t_{12} + t_{13} - t_{14})\langle [1234_+] \rangle \Big)$ |

From the expressions of the couplings given in table 1, one can easily check that these couplings are real. For this, first note that the operator $O$ given in (3) is Hermitian. This implies that its correlators should satisfy relations of the following form:

$$\Big( \langle O(t_1)O(t_2)O(t_3)O(t_4) \rangle \Big)^* = \langle O(t_4)O(t_3)O(t_2)O(t_1) \rangle .$$

(34)

Such relations between the bath's correlators lead to the following conditions on the corresponding cumulants:

$$\langle [1234] \rangle^* = -\langle [1234] \rangle \,, \langle [12_+3_+4_+] \rangle^* = \langle [12_+3_+4_+] \rangle,$$

$$\langle [12_+34] \rangle^* = \langle [12_+34] \rangle \,, \langle [123_+4] \rangle^* = \langle [123_+4] \rangle \,, \langle [1234_+] \rangle^* = \langle [1234_+] \rangle,$$

$$\langle [12_+3_+4] \rangle^* = -\langle [12_+3_+4] \rangle \,, \langle [12_+34_+] \rangle^* = -\langle [12_+34_+] \rangle \,, \langle [123_+4_+] \rangle^* = -\langle [123_+4_+] \rangle .$$

(35)

From these conditions, the reality of the couplings (see table 1) is manifest.

In section 4.5, we will evaluate these couplings (along with the additional OTO couplings) for the qXY model at the high temperature limit of the bath. For now, we would like the reader to just note that the leading order terms in the quadratic and the quartic couplings are $O(\lambda^2)$ and $O(\lambda^4)$ respectively. In the following subsection, we will show that these leading order forms of the couplings enter as parameters in a dual stochastic dynamics.

## 3.5 Duality with a non-linear Langevin dynamics

In this subsection, we show that the quartic Schwinger-Keldysh effective theory of the particle is dual to a classical stochastic theory governed by a non-linear Langevin equation. As mentioned in the introduction, this stochastic dynamics has a structure similar to the one obtained for the cubic effective theory in [30]. Our analysis will be based on the techniques developed by Martin-Siggia-Rose [52], De Dominicis-Peliti [53] and Janssen [54] for obtaining such dualities between quantum mechanical path integrals and stochastic path integrals. We will first propose the form of the dual non-linear Langevin dynamics and then demonstrate its equivalence to the quartic effective theory discussed in the previous subsection.

**The dual non-linear Langevin dynamics:**
Consider a non-linear Langevin equation of the following form:

$$
\begin{aligned}
\mathcal{E}[q, \mathcal{N}] \equiv{}& \ddot{q} + \left(\gamma + \zeta_\gamma^{(2)} \mathcal{N}^2\right)\dot{q} + \left(\overline{\mu}^2 + \zeta_\mu^{(2)} \mathcal{N}^2\right)q + \mathcal{N}\left(\overline{\zeta}_3 - \overline{\zeta}_{3\gamma}\frac{d}{dt}\right)\frac{q^2}{2!} \\
& + \left(\overline{\lambda}_4 - \overline{\lambda}_{4\gamma}\frac{d}{dt}\right)\frac{q^3}{3!} - \langle f^2\rangle\mathcal{N} = 0,
\end{aligned}
\tag{36}
$$

where $\mathcal{N}$ is a noise drawn from a non-Gaussian probability distribution given below

$$
P[\mathcal{N}] \propto \exp\left[-\int dt\left(\frac{\langle f^2\rangle}{2}\mathcal{N}^2 + \frac{Z_I}{2}\dot{\mathcal{N}}^2 + \frac{\zeta_N^{(4)}}{4!}\mathcal{N}^4\right)\right].
\tag{37}
$$

The non-linearities in this dynamics as well as the non-Gaussianity in the noise are fixed by the following parameters: $\zeta_N^{(4)}, \zeta_\gamma^{(2)}, \zeta_\mu^{(2)}, \overline{\zeta}_3, \overline{\zeta}_{3\gamma}, \overline{\lambda}_4, \overline{\lambda}_{4\gamma}$. From equation (33) and table 1, we can see that all these parameters are $O(\lambda^4)$. If we ignore these $O(\lambda^4)$ contributions, then the dynamics satisfies a linear Langevin equation of the following form:

$$
\ddot{q} + \gamma\dot{q} + \overline{\mu}^2 q = \langle f^2\rangle\mathcal{N},
\tag{38}
$$

where the noise is drawn from the Gaussian probability distribution given below

$$
P[\mathcal{N}] \propto \exp\left[-\int dt\left(\frac{\langle f^2\rangle}{2}\mathcal{N}^2 + \frac{Z_I}{2}\dot{\mathcal{N}}^2\right)\right].
\tag{39}
$$

- **Parameters in the linear Langevin dynamics**

  The parameters appearing in the linear Langevin dynamics given in (38) and (39) can be interpreted in the following manner:

  1. $\langle f^2\rangle$ is the strength of an additive noise in the dynamics.

  2. $Z_I$ introduces nonzero correlations between the noise at two different times.

  3. $\overline{\mu}$ is the renormalised frequency.

  4. $\gamma$ is the coefficient of damping.

- **Additional parameters in the non-linear dynamics**

  If we include the contribution of the $O(\lambda^4)$ parameters in the dynamics, then these additional parameters can be interpreted as follows:

  1. $\zeta_\mu^{(2)}$ is a jitter in the renormalised frequency due to the thermal noise.

2. $\zeta_\gamma^{(2)}$ is a jitter in the damping coefficient due to the thermal noise.

3. $\zeta_N^{(4)}$ is the strength of non-Gaussianity in the noise distribution.

4. $\overline{\lambda}_4$ and $\overline{\lambda}_{4\gamma}$ are the strengths of anharmonic terms in the equation of motion.

5. $\overline{\zeta}_3$ and $\overline{\zeta}_{3\gamma}$ are the strengths of anharmonic terms which couple to the noise.

Now, let us demonstrate the duality between this non-linear Langevin dynamics and the quartic effective theory that we introduced earlier.

**Argument for the duality:**

Consider the following stochastic path integral[17] for the non-linear Langevin dynamics given in (36) and (37):

$$\mathcal{Z} = \int [D\mathcal{N}][Dq] e^{\int dt \left( \frac{\zeta_\mu^{(2)} \mathcal{N} q}{\langle f^2 \rangle \delta t} + \frac{\zeta_\gamma^{(2)} \mathcal{N} \dot{q}}{\langle f^2 \rangle \delta t} + \frac{\zeta_N^{(4)} \mathcal{N}^2}{4 \langle f^2 \rangle \delta t} \right)} \delta(\mathcal{E}[q, \mathcal{N}]) P[\mathcal{N}] . \qquad (40)$$

Here, $\delta t$ is a UV-regulator for 2-point functions of $\mathcal{N}$, and $\left[ \int dt \left( \frac{\zeta_\mu^{(2)} \mathcal{N} q}{\langle f^2 \rangle \delta t} + \frac{\zeta_\gamma^{(2)} \mathcal{N} \dot{q}}{\langle f^2 \rangle \delta t} + \frac{\zeta_N^{(4)} \mathcal{N}^2}{4 \langle f^2 \rangle \delta t} \right) \right]$ is a counter-term introduced to cancel the regulator-dependent contributions arising from loop integrals of $\mathcal{N}$.

Notice that q is a dummy variable in the above path integral. We choose to relabel this variable as $q_a$. In addition, we introduce an auxiliary variable $q_d$ which replaces the delta function for the equation of motion by an integral as shown below

$$\mathcal{Z} = \int [D\mathcal{N}][Dq_a][Dq_d] e^{\int dt \left( \frac{\zeta_\mu^{(2)} \mathcal{N} q_a}{\langle f^2 \rangle \delta t} + \frac{\zeta_\gamma^{(2)} \mathcal{N} \dot{q}_a}{\langle f^2 \rangle \delta t} + \frac{\zeta_N^{(4)} \mathcal{N}^2}{4 \langle f^2 \rangle \delta t} \right)} e^{-i \int dt \mathcal{E}[q_a, \mathcal{N}] q_d} P[\mathcal{N}]$$

$$= \int [D\mathcal{N}][Dq_a][Dq_d] \exp\Big[ i \int dt \Big\{ -i \frac{\zeta_\mu^{(2)} \mathcal{N} q_a}{\langle f^2 \rangle \delta t} - i \frac{\zeta_\gamma^{(2)} \mathcal{N} \dot{q}_a}{\langle f^2 \rangle \delta t} - i \frac{\zeta_N^{(4)} \mathcal{N}^2}{4 \langle f^2 \rangle \delta t} - q_d \ddot{q}_a$$

$$- q_d \big( \gamma + \zeta_\gamma^{(2)} \mathcal{N}^2 \big) \dot{q}_a - q_d \big( \overline{\mu}^2 + \zeta_\mu^{(2)} \mathcal{N}^2 \big) q_a \qquad (41)$$

$$- \mathcal{N} q_d \Big( \overline{\zeta}_3 - \overline{\zeta}_{3\gamma} \frac{d}{dt} \Big) \frac{q_a^2}{2!} - q_d \Big( \overline{\lambda}_4 - \overline{\lambda}_{4\gamma} \frac{d}{dt} \Big) \frac{q_a^3}{3!}$$

$$+ \langle f^2 \rangle \mathcal{N} q_d + i \frac{\langle f^2 \rangle}{2} \mathcal{N}^2 + i \frac{Z_I}{2} \dot{\mathcal{N}}^2 + i \frac{\zeta_N^{(4)}}{4!} \mathcal{N}^4 \Big\} \Big] .$$

Now, let us introduce the following shift in the noise variable appearing in the above path integral:

$$\mathcal{N} \to \mathcal{N} + i q_d . \qquad (42)$$

Integrating out this shifted noise variable leads to a residual path integral over $q_a$ and $q_d$. In the action of this residual path integral, we retain all the quadratic terms up to $O(\lambda^2)$, and all the quartic terms up to $O(\lambda^4)$.[18] Up to this approximation, the residual path integral has the

---

[17]We ignore the Jacobian $\det\left[ \frac{\delta \mathcal{E}[q(t), \mathcal{N}(t)]}{\delta q(t')} \right]$ in the path integral as it does not contribute to the quadratic and quartic terms that we eventually get in (47) upto leading orders in $\lambda$.

[18]While integrating out the noise, we consider the terms associated with $Z_I$ and all the $O(\lambda^4)$ parameters in the action to be small corrections over the term associated with $\langle f^2 \rangle$. Then the 2-point function of the noise reduces to

$$\langle \mathcal{N}(t_1) \mathcal{N}(t_2) \rangle = \frac{1}{\langle f^2 \rangle} \delta(t_1 - t_2) + (\text{perturbative corrections}). \qquad (43)$$

We ignore the perturbative corrections and regulate the delta function by the UV regulator $\delta t$ that we introduced

following form:

$$
\begin{aligned}
\mathcal{Z} = C \int [Dq_a][Dq_d] \exp\Big[ i \int dt \Big\{ & -q_d \ddot{q}_a - \gamma q_d \dot{q}_a - \overline{\mu}^2 q_d q_a - i\frac{Z_I}{2}\dot{q}_d{}^2 + i\frac{\langle f^2 \rangle}{2}q_d^2 \\
& + i\frac{\zeta_N^{(4)}}{4!}q_d^4 + \zeta_\mu^{(2)}q_d^3 q_a + \zeta_\gamma^{(2)}q_d^3\dot{q}_a - i\overline{\zeta}_3\frac{q_d^2 q_a^2}{2!} \\
& + i\overline{\zeta}_{3\gamma}q_d^2 q_a \dot{q}_a - \overline{\lambda}_4\frac{q_d q_a^3}{3!} + \overline{\lambda}_{4\gamma}\frac{q_d q_a^2 \dot{q}_a}{2!} \Big\} \Big],
\end{aligned}
\tag{45}
$$

where C is a constant given by

$$
C = \int [D\mathcal{N}]e^{\int dt \frac{\zeta_N^{(4)}\mathcal{N}^2}{4\langle f^2 \rangle \delta t}} e^{-\int dt \left( \frac{\langle f^2 \rangle}{2}\mathcal{N}^2 + \frac{Z_I}{2}\dot{\mathcal{N}}^2 + \frac{\zeta_N^{(4)}}{4!}\mathcal{N}^4 \right)}.
\tag{46}
$$

Integrating by parts the first term and the last term in the action of the above path integral, we get

$$
\begin{aligned}
\mathcal{Z} = C \int [Dq_a][Dq_d] \exp\Big[ i \int dt \Big\{ & \dot{q}_d \dot{q}_a - i\frac{Z_I}{2}\dot{q}_d{}^2 - \gamma q_d \dot{q}_a + i\frac{\langle f^2 \rangle}{2}q_d^2 - \overline{\mu}^2 q_d q_a \\
& + i\frac{\zeta_N^{(4)}}{4!}q_d^4 + \zeta_\mu^{(2)}q_d^3 q_a + \zeta_\gamma^{(2)}q_d^3\dot{q}_a - i\overline{\zeta}_3\frac{q_d^2 q_a^2}{2!} \\
& + i\overline{\zeta}_{3\gamma}q_d^2 q_a \dot{q}_a - \overline{\lambda}_4\frac{q_d q_a^3}{3!} - \overline{\lambda}_{4\gamma}\frac{\dot{q}_d q_a^3}{3!} \Big\} \Big].
\end{aligned}
\tag{47}
$$

Now, notice that the action in the above expression is exactly the Schwinger-Keldysh effective action given in (28) and (29) under the following identification:

$$
q_a \equiv \frac{q_1 - q_2}{2}, \; q_d \equiv q_1 + q_2 .
\tag{48}
$$

This basis $\{q_a, q_d\}$ for the Schwinger-Keldysh degrees of freedom is commonly known as the Keldysh basis [3]. The Schwinger-Keldysh effective Lagrangian of the particle in this basis is given by

$$
\boxed{
\begin{aligned}
L_{\text{SK,1-PI}} = & \dot{q}_d \dot{q}_a - \frac{i}{2}Z_I \dot{q}_d^2 - \overline{\mu}^2 q_d q_a + \frac{i}{2}\langle f^2 \rangle q_d^2 - \gamma q_d \dot{q}_a \\
& + \frac{i\zeta_N^{(4)}}{4!}q_d^4 + \zeta_\mu^{(2)}q_d^3 q_a - \frac{i\overline{\zeta}_3}{2!}q_d^2 q_a^2 - \frac{\overline{\lambda}_4}{3!}q_d q_a^3 \\
& + \zeta_\gamma^{(2)}q_d^3 \dot{q}_a - \frac{\overline{\lambda}_{4\gamma}}{3!}\dot{q}_d q_a^3 + i\overline{\zeta}_{3\gamma}q_d^2 q_a \dot{q}_a .
\end{aligned}
}
\tag{49}
$$

Therefore, one can express the stochastic path integral in terms of the Schwinger-Keldysh effective action as shown below

$$
\boxed{\mathcal{Z} = C \int [Dq_a][Dq_d]e^{i\int dt L_{\text{SK,1-PI}}} .}
\tag{50}
$$

---

earlier. Then the equal-time 2-point correlator of $\mathcal{N}$ reduces to

$$
\langle \mathcal{N}^2 \rangle = \frac{1}{\langle f^2 \rangle \delta t}.
\tag{44}
$$

The contribution of $\langle \mathcal{N}^2 \rangle$ to the action of the residual path integral exactly cancels the contribution from the counter-term.

This concludes our argument for the duality between the quartic effective theory and the non-linear Langevin dynamics. We refer the reader to [6] for a more detailed discussion on such dualities between stochastic and Schwinger-Keldysh path integrals.

**A brief comment on the sign of $\zeta_N^{(4)}$:**

In section 4.5, we will see that, for the $qXY$ model, the value of $\zeta_N^{(4)}$ is negative. This may raise concern about the validity of the probability distribution given in (37) since it diverges when $|\mathcal{N}| \to \infty$. However, we would like to remind the reader that we have done a perturbative analysis here and ignored all possible corrections to the probability distribution beyond the quartic order. Such a perturbative analysis is insufficient to determine the behaviour of the probabilty density at large values of $\mathcal{N}$.

### 3.6 Limitation of the Schwinger-Keldysh effective theory

The Schwinger-Keldysh effective theory or the dual non-linear Langevin dynamics developed in this section suffers from the following limitations:

1. It allows one to compute only correlators of the particle which can be obtained from path integrals on the Schwinger-Keldysh contour shown in figure 2.

2. The effective couplings receive contributions only from similar Schwinger-Keldysh correlators of the bath.

However, it is possible to consider more general correlators of the particle which cannot be obtained from path integrals on the Schwinger-Keldysh contour. For example, consider the correlator

$$\langle q(t)q(0)q(t)q(0)\rangle \equiv \text{Tr}[\rho_0 q(t)q(0)q(t)q(0)], \tag{51}$$

where $t_f > t > 0 > t_0$. Starting from the initial density matrix in the above expression, one has to go forward and backward in time twice to include all the insertions. To get a path integral representation for such correlators, one needs to consider a contour with two time folds [32, 33] as shown in figure 3. The positions of the insertions that would give the correlator in (51) are indicated in this diagram by the crosses. These insertions are contour-ordered according to the arrow indicated in the diagram.

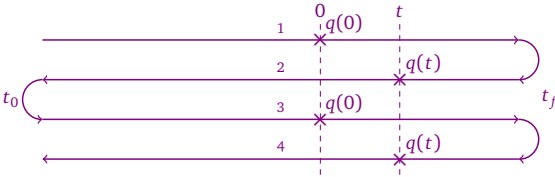

Figure 3: Contour-ordered correlator for $\langle q(t)q(0)q(t)q(0)\rangle$

Correlators which can only be obtained by putting insertions on such a contour with multiple time-folds are known as out-of-time-order correlators (OTOCs). The construction of an effective theory for computing such OTOCs requires incorporating the effects of similar OTOCs of the bath [45]. As we mentioned in the introduction, such thermal OTOCs in a bath have emerged as very important diagnostic measures of chaos, thermalisation, many-body localisation, etc. [34–36, 38, 41–44]. Moreover, it is essential to include these OTOCs of the bath in the study of the analytic properties of its thermal correlators [9, 30]. These analytic properties of the bath's correlators impose nontrivial constraints [30] on the effective couplings of the Brownian particle.

In the next section, we will first extend the effective theory to include the particle's OTOCs. Then we will derive the constraints imposed on the quartic couplings in this OTO effective dynamics by the analytic properties of the bath's correlators.

# 4 Extension to the effective theory for OTO correlators

In this section, we will extend the effective theory of the particle to one that

a) allows computation of the particle's OTOCs, and

b) takes into account the contributions of the bath's OTOCs.

In particular, we will see that the effective couplings in this extended framework receive contributions from the OTOCs of the operator $O$ that couples to the particle. We will show that some of these OTO correlators of the bath are related to Schwinger-Keldysh correlators due to the following two reasons:

1. microscopic reversibility in the bath's dynamics,

2. thermality of the bath.

These relations between the bath's correlators, in turn, impose certain constraints on the quartic effective couplings. Following the methods illustrated in [30], we will derive these constraints, and show that they lead to generalisations of the Onsager reciprocal relations [46,47] and the fluctuation-dissipation relation [25,30,49]. Finally, we will provide the values of the effective couplings for the qXY model introduced in section 2 and check that they satisfy all the constraints.

## 4.1 Generalised influence phase of the particle

At the end of the last section, we saw that a path integral representation for the OTOCs requires us to introduce a contour with two time-folds as shown in figure 3. Then following the strategy developed for obtaining the Schwinger-Keldysh effective dynamics, here we will have to consider a copy of the microscopic degrees of freedom for each of the four legs in this contour: $\{q_1, X_1, Y_1\}$, $\{q_2, X_2, Y_2\}$, $\{q_3, X_3, Y_3\}$ and $\{q_4, X_4, Y_4\}$.

The Lagrangian of the combined system in this generalised Schwinger-Keldysh path integral is given by[19]

$$L_{\text{GSK}} = L[q_1, X_1, Y_1] - L[-q_2, X_2, Y_2] + L[q_3, X_3, Y_3] - L[-q_4, X_4, Y_4]. \tag{52}$$

While computing OTO correlators of the particle from path integrals with the above Lagrangian, we can first integrate out the bath's degrees of freedom. This would give us a generalisation of the influence phase which encodes the effect of the OTOCs of the bath on the particle's effective dynamics. This generalised influence phase [45] is given by

$$W_{GSK} = \sum_{n=1}^{\infty} \lambda^n W_{GSK}^{(n)}, \tag{53}$$

where

$$W_{GSK}^{(n)} = i^{n-1} \sum_{i_1, \cdots, i_n=1}^{4} \int_{t_0}^{t_f} dt_1 \int_{t_0}^{t_1} dt_2 \cdots \int_{t_0}^{t_{n-1}} dt_n \, \langle \mathcal{T}_C O_{i_1}(t_1) \cdots O_{i_n}(t_n) \rangle_c \, q_{i_1}(t_1) \cdots q_{i_n}(t_n). \tag{54}$$

---

[19]Here we again put extra minus signs while defining the q's on the even legs. This is consistent with the convention followed in [45] and [30].

As in case of the influence phase, one can expand the q's in (54) at the times $t_2, \cdots, t_n$ about $t_1$ to get an approximately local form for the generalised influence phase. The quadratic and the quartic terms in this approximately local form are given in appendix C.

The approximately local generalised influence phase can be used to compute the OTO correlators of the particle. But just like the influence phase, it suffers from the problem of having point-split regulators to preserve the original time-ordering in the non-local form. To avoid this complication, we will follow the method introduced in [45] to construct a local 1-PI effective action of the particle on the 2-fold contour. As we will see next, this OTO 1-PI effective action is a straightforward extension of the Schwinger-Keldysh effective action discussed in the previous section.

## 4.2 Out of time ordered 1-PI effective action of the particle

The 1-PI effective action for OTOCs has to satisfy some constraints which are based on similar principles as those mentioned in section 3.4. These constraints were first discussed in [45] and later used in [30] to study the quadratic and cubic terms in the OTO effective theory of the qXY model. We summarise these constraints below:

- **Collapse rules:**

  The 1-PI effective action should become independent of $\tilde{q}$ under any of the following identifications:

    1. $q_1 = -q_2 = \tilde{q}$,

    2. $q_2 = -q_3 = \tilde{q}$,

    3. $q_3 = -q_4 = \tilde{q}$.

  Moreover, under any of these collapses, the OTO effective action should reduce to the Schwinger-Keldysh effective action introduced earlier.

- **Reality conditions:**

  The effective action should become its own negative under complex conjugation of the effective couplings along with the following exchanges:

$$q_1 \longleftrightarrow -q_4, \; q_2 \longleftrightarrow -q_3 \; . \tag{55}$$

In addition, the 1-PI effective action on the 2-fold contour should be invariant under

$$q_1 \to -q_1, \; q_2 \to -q_2, \; q_3 \to -q_3, \; q_4 \to -q_4 \tag{56}$$

to respect the symmetries given in (25).

The most general 1-PI effective action which is consistent with these conditions can be expanded as

$$L_{\text{1-PI}} = L_{\text{1-PI}}^{(2)} + L_{\text{1-PI}}^{(4)} + \cdots , \tag{57}$$

where $L_{\text{1-PI}}^{(2)}$ and $L_{\text{1-PI}}^{(4)}$ are the quadratic and the quartic terms in the action respectively.

**Quadratic terms:** The quadratic terms can be obtained by extending their Schwinger-Keldysh (SK) counterparts given in (28) [45]. We provide the form of these terms below:

$$
\begin{aligned}
L_{1\text{-PI}}^{(2)} =& \frac{1}{2}(\dot{q}_1^2 - \dot{q}_2^2 + \dot{q}_3^2 - \dot{q}_4^2) - \frac{\overline{\mu}^2}{2}(q_1^2 - q_2^2 + q_3^2 - q_4^2) \\
& - \frac{\gamma}{2}\Big[(q_1 + q_2)(\dot{q}_1 - \dot{q}_2 - \dot{q}_3 - \dot{q}_4) + (q_3 + q_4)(\dot{q}_1 + \dot{q}_2 + \dot{q}_3 - \dot{q}_4)\Big] \\
& + \frac{i\langle f^2\rangle}{2}(q_1 + q_2 + q_3 + q_4)^2 - \frac{iZ_I}{2}(\dot{q}_1 + \dot{q}_2 + \dot{q}_3 + \dot{q}_4)^2 .
\end{aligned}
\tag{58}
$$

One can easily check that these quadratic terms reduce to the corresponding terms in the SK effective action under any of the collapses mentioned above.

**Quartic terms:** The quartic terms can be split into two parts [45]:

1. terms which reduce to their SK counterparts given in (29) under any of the collapses,

2. terms which go to zero under these collapses.

Accordingly, the quartic effective action can be written as

$$
L_{1\text{-PI}}^{(4)} = L_{1\text{-PI,SK}}^{(4)} + L_{1\text{-PI,OTO}}^{(4)} ,
\tag{59}
$$

where $L_{1\text{-PI,SK}}^{(4)}$ and $L_{1\text{-PI,OTO}}^{(4)}$ are the two sets of terms mentioned above.

**Extension of quartic terms in Schwinger-Keldysh effective theory:** The extension of the quartic terms in the SK effective theory can be further divided into different sets according to the number of time derivatives in them. Here, we keep terms with up to a single derivative acting on the q's. Then $L_{1\text{-PI,SK}}^{(4)}$ can be decomposed as

$$
L_{1\text{-PI,SK}}^{(4)} = L_{1\text{-PI,SK}}^{(4,0)} + L_{1\text{-PI,SK}}^{(4,1)} + \cdots ,
\tag{60}
$$

where $L_{1\text{-PI,SK}}^{(4,0)}$ and $L_{1\text{-PI,SK}}^{(4,1)}$ are terms without any derivative and terms with a single derivative respectively. We give the forms of these terms below:

$$
\begin{aligned}
& L_{1\text{-PI,SK}}^{(4,0)} \\
&= -\frac{\overline{\lambda}_4}{48}\Big[(q_1 + q_2)(q_1 - q_2 - q_3 - q_4)^3 + (q_3 + q_4)(q_1 + q_2 + q_3 - q_4)^3\Big] \\
&\quad + \frac{\zeta_\mu^{(2)}}{2}(q_1 + q_2 + q_3 + q_4)^2(q_1^2 - q_2^2 + q_3^2 - q_4^2) - \frac{i\overline{\zeta}_3}{8}(q_1^2 - q_2^2 + q_3^2 - q_4^2)^2 \\
&\quad + \frac{i\zeta_N^{(4)}}{24}(q_1 + q_2 + q_3 + q_4)^4 ,
\end{aligned}
\tag{61}
$$

$$
\begin{aligned}
L^{(4,1)}_{\text{1-PI,SK}} \\
= -\frac{\overline{\lambda}_{4\gamma}}{48}&\Big[(\dot{q}_1 + \dot{q}_2)(q_1 - q_2 - q_3 - q_4)^3 + (\dot{q}_3 + \dot{q}_4)(q_1 + q_2 + q_3 - q_4)^3\Big] \\
+ \frac{\zeta^{(2)}_\gamma}{2}&(q_1 + q_2 + q_3 + q_4)^2\Big[(q_1 + q_2)(\dot{q}_1 - \dot{q}_2 - \dot{q}_3 - \dot{q}_4) + (q_3 + q_4)(\dot{q}_1 + \dot{q}_2 + \dot{q}_3 - \dot{q}_4)\Big] \\
+ \frac{i\overline{\zeta}_{3\gamma}}{4}&(q_1 + q_2 + q_3 + q_4)\Big[(q_1 + q_2)(q_1 - q_2 - q_3 - q_4)(\dot{q}_1 - \dot{q}_2 - \dot{q}_3 - \dot{q}_4) \\
& \qquad\qquad + (q_3 + q_4)(q_1 + q_2 + q_3 - q_4)(\dot{q}_1 + \dot{q}_2 + \dot{q}_3 - \dot{q}_4)\Big].
\end{aligned}
$$

(62)

**Additional OTO quartic terms:** The additional OTO quartic terms which vanish under any of the collapses can also be split into terms with different number of derivatives acting on the q's. As in case of the extension of the SK effective action terms, $L^{(4)}_{\text{1-PI,OTO}}$ can be decomposed as

$$
L^{(4)}_{\text{1-PI,OTO}} = L^{(4,0)}_{\text{1-PI,OTO}} + L^{(4,1)}_{\text{1-PI,OTO}} + \cdots ,
$$

(63)

where $L^{(4,0)}_{\text{1-PI,OTO}}$ and $L^{(4,1)}_{\text{1-PI,OTO}}$ are terms without any derivative and terms with a single derivative respectively. The forms of these terms are given below:

$$
\begin{aligned}
L^{(4,0)}_{\text{1-PI,OTO}} \\
= (q_1 + q_2)(q_2 + q_3)(q_3 + q_4)\Big[A_1(q_1 + q_2) + A_2(q_3 + q_4) + A_3(q_1 - q_4) + A_4(q_2 + q_3)\Big],
\end{aligned}
$$

(64)

$$
\begin{aligned}
L^{(4,1)}_{\text{1-PI,OTO}} = &(q_1 + q_2)(q_2 + q_3)(\dot{q}_3 + \dot{q}_4)\Big[B_1(q_1 - q_4) + B_2(q_1 + q_2) + B_3(q_2 + q_3)\Big] \\
& + (q_1 + q_2)(\dot{q}_2 + \dot{q}_3)(q_3 + q_4)\Big[B_4(q_1 - q_4) + B_5(q_1 + q_2) + B_6(q_3 + q_4)\Big] \\
& + (\dot{q}_1 + \dot{q}_2)(q_2 + q_3)(q_3 + q_4)\Big[B_7(q_1 - q_4) + B_8(q_3 + q_4) + B_9(q_2 + q_3)\Big].
\end{aligned}
$$

(65)

The reality conditions impose the following constraints on the quartic OTO couplings:

$$
\begin{aligned}
&A_1 = -A_2^*,\ A_3 = A_3^*,\ A_4 = -A_4^*, \\
&B_1 = B_7^*,\ B_2 = -B_8^*,\ B_3 = -B_9^*,\ B_4 = B_4^*,\ B_5 = -B_6^*.
\end{aligned}
$$

(66)

These additional OTO terms will not contribute to any correlator on the 2-fold contour that can also be obtained from the Schwinger-Keldysh contour. This is ensured by the collapse rules mentioned above. However, they are essential for computing OTOCs of the particle. The couplings appearing in these OTO terms, in turn, receive contributions from the OTOCs of the bath. As we will discuss next, some of these OTOCs of the bath are related to Schwinger-Keldysh correlators due to the microscopic reversibility in the dynamics of the bath. We will show that such relations impose constraints on the quartic couplings in the effective action which can be intepreted as generalisations of the Onsager reciprocal relations [46, 47].

### 4.3 Generalised Onsager relations

In the qXY model, the bath's dynamics (excluding the perturbation by the particle) has a symmetry under time-reversal:

$$\mathbf{T}X^{(i)}(t)\mathbf{T}^{\dagger} = X^{(i)}(-t), \ \mathbf{T}Y^{(j)}(t)\mathbf{T}^{\dagger} = Y^{(j)}(-t) . \tag{67}$$

Here, $\mathbf{T}$ is an anti-linear and anti-unitary operator[20] which commutes with the bath's Hamiltonian i.e.

$$[\mathbf{T}, H_B] = 0. \tag{68}$$

Moreover, the bath operator $O \equiv \sum_{i,j} g_{xy,ij} X^{(i)} Y^{(j)}$ that couples to the particle has an even parity under time-reversal which implies

$$\mathbf{T}O(t)\mathbf{T}^{\dagger} = O(-t). \tag{69}$$

Such a microscopic reversibility in the bath's dynamics introduces certain relations between the couplings in the effective theory of the particle. These relations were first discovered by Onsager [46, 47] and later extended by Casimir [48] in the context of the quadratic effective theory of a system with multiple degrees of freedom. Let us denote these degrees of freedom by $\{q_A\}$, and assume that $q_A$ couples to the bath operator $O_A$ with parity $\eta_A$ under time-reversal. As a concrete example, one can take these degrees of freedom to be the coordinates of a Brownian particle moving in multiple dimensions. In such a setup, the frequency, the damping and the noise coefficients in the particle's effective dynamics will be replaced by matrices $\overline{\mu}^2_{AB}$, $\gamma_{AB}$ and $\langle f^2 \rangle_{AB}$. The first index (A) in these coefficients indicates the degree of freedom in whose equation of motion they appear. The second index (B) indicates the degree of freedom with which these coefficients get multiplied in the equation of motion of $q_A$. For these coefficients, Onsager and Casimir derived reciprocal relations of the following form

$$\overline{\mu}^2_{AB} = \eta_A \eta_B \overline{\mu}^2_{BA}, \ \gamma_{AB} = \eta_A \eta_B \gamma_{BA}, \ \langle f^2 \rangle_{AB} = \eta_A \eta_B \langle f^2 \rangle_{BA} . \tag{70}$$

For a system with a single degree of freedom, as in case of the Brownian particle moving in 1 dimension, the Onsager-Casimir relations are trivially satisfied by the quadratic couplings. So, at the level of the quadratic effective theory, there is no constraint imposed by the microscopic reversibility in the bath.

However, an extension of such relations was found in [30] for the cubic OTO effective theory of the Brownian particle. It was observed that, when there is microscopic reversibility in the bath, all the cubic OTO couplings are related to the cubic Schwinger-Keldysh couplings. These relations between the effective couplings arise from certain relations between the 3-point correlators of the bath which are rooted in its microscopic reversibility. Here, we will first show that similar relations exist between the 4-point correlators of the bath. Then we will discuss the constraints imposed on the quartic effective couplings of the particle due to these relations between the 4-point functions of the bath.

**Relations between the bath's correlators due to microscopic reversibility:**

The microscopic reversibility in the bath leads to the following kind of relations between the 4-point correlators of $O$:

$$\langle O(t_1)O(t_2)O(t_3)O(t_4) \rangle = \left( \langle \mathbf{T}O(t_1)\mathbf{T}^{\dagger}\mathbf{T}O(t_2)\mathbf{T}^{\dagger}\mathbf{T}O(t_3)\mathbf{T}^{\dagger}\mathbf{T}O(t_4)\mathbf{T}^{\dagger} \rangle \right)^{*} . \tag{71}$$

Using the transformation of $O(t)$ under time-reversal (see equation (69)), we get

$$\langle O(t_1)O(t_2)O(t_3)O(t_4) \rangle = \left( \langle O(-t_1)O(-t_2)O(-t_3)O(-t_4) \rangle \right)^{*} . \tag{72}$$

---

[20]We refer the reader to [55] for a discussion on these properties of the time-reversal operator.

Now, due to the Hermiticity of the operator $O$, the above relation reduces to

$$\langle O(t_1)O(t_2)O(t_3)O(t_4)\rangle = \langle O(-t_4)O(-t_3)O(-t_2)O(-t_1)\rangle . \tag{73}$$

Note that such relations can connect two OTO correlators as in the following example:

$$\langle O(t)O(0)O(t)O(0)\rangle = \langle O(0)O(-t)O(0)O(-t)\rangle . \tag{74}$$

Moreover, they can also connect an OTO correlator to a Schwinger-Keldysh correlator. For example, consider the relation

$$\langle O(t)O(0)O(0)O(t)\rangle = \langle O(-t)O(0)O(0)O(-t)\rangle , \tag{75}$$

where $t > 0$. The correlator on the left hand side of the above equation is an OTOC which can be obtained by putting insertions on a 2-fold contour as shown in figure 4. The correlator on

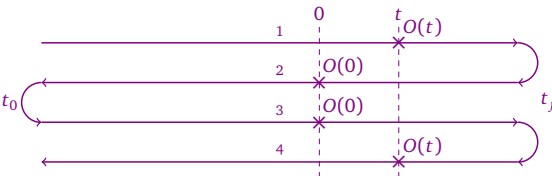

Figure 4: Contour-ordered correlator for $\langle O(t)O(0)O(0)O(t)\rangle$ where $t > 0$

the right-hand side of (75), however, can be obtained from a path integral on the Schwinger-Keldysh contour as shown in figure 5.

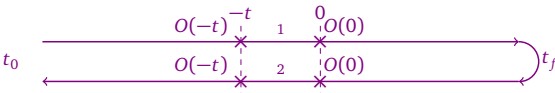

Figure 5: Contour-ordered correlator for $\langle O(-t)O(0)O(0)O(-t)\rangle$ where $t > 0$

Such relations between the bath's 4-point correlators can introduce two kinds of constraints on the quartic effective couplings of the particle:

1. They can lead to a relation between two different OTO couplings.

2. They can also relate an OTO coupling to a Schwinger-Keldysh coupling.

We will now derive these constraints on the effective couplings of the particle.

**Constraints on the quartic effective couplings:**

To analyse the constraints on the effective couplings, we find it convenient to re-express the quartic OTO couplings in terms of some new real parameters as shown below:

$$\boxed{\begin{aligned}
A_1 &= -A_2^* = \frac{1}{12}\Big[(-\overline{\lambda}_4 + \widetilde{\kappa}_4) - 6i(\varrho_4 - \widetilde{\varrho}_4)\Big] , \\
A_3 &= A_3^* = \kappa_4 , \\
A_4 &= -A_4^* = \frac{i}{2}(\varrho_4 + \widetilde{\varrho}_4) .
\end{aligned}} \tag{76}$$

$$
\begin{aligned}
B_1 = B_7^* &= \frac{1}{4}\Big[(2\kappa_{4\gamma}^{II} + 4\kappa_{4\gamma}^{III} + \widetilde{\kappa}_{4\gamma}^{II}) + i(\varrho_{4\gamma}^{II} + \widetilde{\varrho}_{4\gamma}^{I} - \widetilde{\varrho}_{4\gamma}^{II})\Big], \\
B_2 = -B_8^* &= \frac{1}{16}\Big[(-\overline{\lambda}_{4\gamma} + 24\zeta_{\gamma}^{(2)} + 12\kappa_{4\gamma}^{I} - 4\widetilde{\kappa}_{4\gamma}^{I} - 6\widetilde{\kappa}_{4\gamma}^{II}) \\
&\quad + i(4\overline{\zeta}_{3\gamma} - 2\varrho_{4\gamma}^{I} + 2\varrho_{4\gamma}^{II} - 6\widetilde{\varrho}_{4\gamma}^{I} + 2\widetilde{\varrho}_{4\gamma}^{II})\Big], \\
B_3 = -B_9^* &= \frac{1}{4}\Big[(-2\kappa_{4\gamma}^{II} + \widetilde{\kappa}_{4\gamma}^{II}) + i\varrho_{4\gamma}^{I}\Big], \\
B_4 = B_4^* &= \frac{1}{2}\Big(2\kappa_{4\gamma}^{II} + \widetilde{\kappa}_{4\gamma}^{II}\Big), \\
B_5 = -B_6^* &= \frac{1}{16}\Big[(\overline{\lambda}_{4\gamma} + 8\zeta_{\gamma}^{(2)} + 4\kappa_{4\gamma}^{I} + 4\widetilde{\kappa}_{4\gamma}^{I} - 2\widetilde{\kappa}_{4\gamma}^{II}) \\
&\quad + i(-4\overline{\zeta}_{3\gamma} + 2\varrho_{4\gamma}^{I} + 6\varrho_{4\gamma}^{II} - 2\widetilde{\varrho}_{4\gamma}^{I} + 6\widetilde{\varrho}_{4\gamma}^{II})\Big].
\end{aligned}
\tag{77}
$$

These new parameters are chosen so that they have definite parities under time-reversal (see the discussion below). The presence or absence of tilde over any coupling indicates whether it has odd or even parity respectively. The symbols $\kappa$ and $\varrho$ are used to represent the couplings which get multiplied to real and imaginary terms in the effective action respectively. The subscript $\gamma$ is introduced to distinguish the couplings corresponding to the single derivative terms in the action.

One can determine the dependence of these OTO couplings on the cumulants of the operator $O$ by comparing the particle's 4-point OTOCs obtained from the generalised influence phase with those computed using the OTO 1-PI effective action. We provide the expressions of these couplings in terms of the bath's OTO cumulants in appendix C.

For the purpose of analysing the constraints on the effective couplings, we find it convenient to convert their expressions (see (33),(129) , and tables 1 and 6) into integrals over a frequency domain where the integrands are determined by some spectral functions of the bath. These spectral functions are the Fourier transforms of the 4-point cumulants of $O(t)$ as defined by the relations given in (79), (80) and (81). In these relations, we define the measure for the integral over the frequencies as

$$
\int \frac{d^4\omega}{(2\pi)^4} \equiv \int_{-\infty}^{\infty} \frac{d\omega_1}{2\pi} \int_{-\infty}^{\infty} \frac{d\omega_2}{2\pi} \int_{-\infty}^{\infty} \frac{d\omega_3}{2\pi} \int_{-\infty}^{\infty} \frac{d\omega_4}{2\pi}.
\tag{78}
$$

- **Spectral functions for Wightman correlators:**

$$
\begin{aligned}
\int \frac{d^4\omega}{(2\pi)^4} \rho\langle 1234\rangle \, e^{-i(\omega_1 t_1 + \omega_2 t_2 + \omega_3 t_3 + \omega_4 t_4)} &\equiv \lambda^4 \langle O(t_1)O(t_2)O(t_3)O(t_4)\rangle_c, \\
\int \frac{d^4\omega}{(2\pi)^4} \rho\langle 4321\rangle \, e^{-i(\omega_1 t_1 + \omega_2 t_2 + \omega_3 t_3 + \omega_4 t_4)} &\equiv \lambda^4 \langle O(t_4)O(t_3)O(t_2)O(t_1)\rangle_c, \text{ etc.}
\end{aligned}
\tag{79}
$$

- **Spectral functions for single-nested (anti-)commutators:**

$$\int \frac{d^4\omega}{(2\pi)^4} \rho[1234] \, e^{-i(\omega_1 t_1 + \omega_2 t_2 + \omega_3 t_3 + \omega_4 t_4)} \equiv \lambda^4 \langle [[[O(t_1), O(t_2)], O(t_3)], O(t_4)] \rangle_c,$$

$$\int \frac{d^4\omega}{(2\pi)^4} \rho[12_+3_+4_+] e^{-i(\omega_1 t_1 + \omega_2 t_2 + \omega_3 t_3 + \omega_4 t_4)} \equiv \lambda^4 \langle \{\{\{O(t_1), O(t_2)\}, O(t_3)\}, O(t_4)\} \rangle_c,$$

$$\int \frac{d^4\omega}{(2\pi)^4} \rho[1234_+] \, e^{-i(\omega_1 t_1 + \omega_2 t_2 + \omega_3 t_3 + \omega_4 t_4)} \equiv \lambda^4 \langle \{[[O(t_1), O(t_2)], O(t_3)], O(t_4)\} \rangle_c,$$

$$\int \frac{d^4\omega}{(2\pi)^4} \rho[123_+4] \, e^{-i(\omega_1 t_1 + \omega_2 t_2 + \omega_3 t_3 + \omega_4 t_4)} \equiv \lambda^4 \langle [\{[O(t_1), O(t_2)], O(t_3)\}, O(t_4)] \rangle_c,$$

$$\int \frac{d^4\omega}{(2\pi)^4} \rho[12_+34] \, e^{-i(\omega_1 t_1 + \omega_2 t_2 + \omega_3 t_3 + \omega_4 t_4)} \equiv \lambda^4 \langle [[\{O(t_1), O(t_2)\}, O(t_3)], O(t_4)] \rangle_c, \text{ etc.}$$

$$(80)$$

- **Spectral functions for double (anti-)commutators:**

$$\int \frac{d^4\omega}{(2\pi)^4} \rho[12][34] \, e^{-i(\omega_1 t_1 + \omega_2 t_2 + \omega_3 t_3 + \omega_4 t_4)} \equiv \lambda^4 \langle [O(t_1), O(t_2)][O(t_3), O(t_4)] \rangle_c$$

$$\int \frac{d^4\omega}{(2\pi)^4} \rho[12_+][34_+] \, e^{-i(\omega_1 t_1 + \omega_2 t_2 + \omega_3 t_3 + \omega_4 t_4)} \equiv \lambda^4 \langle \{O(t_1), O(t_2)\}\{O(t_3), O(t_4)\} \rangle_c,$$

$$\int \frac{d^4\omega}{(2\pi)^4} \rho[12_+][34] \, e^{-i(\omega_1 t_1 + \omega_2 t_2 + \omega_3 t_3 + \omega_4 t_4)} \equiv \lambda^4 \langle \{O(t_1), O(t_2)\}[O(t_3), O(t_4)] \rangle_c$$

$$\int \frac{d^4\omega}{(2\pi)^4} \rho[12][34_+] \, e^{-i(\omega_1 t_1 + \omega_2 t_2 + \omega_3 t_3 + \omega_4 t_4)} \equiv \lambda^4 \langle [O(t_1), O(t_2)]\{O(t_3), O(t_4)\} \rangle_c, \text{ etc.}$$

$$(81)$$

Any quartic coupling $g$ can be expressed as an integral of these spectral functions in the following manner:

$$g = \int_{\mathcal{C}_4} \widetilde{\mathcal{I}}[g] + O(\lambda^6). \tag{82}$$

The domain of integration $\mathcal{C}_4$ is given by[21]

$$\int_{\mathcal{C}_4} \equiv \int_{-\infty - i\epsilon_1}^{\infty - i\epsilon_1} \frac{d\omega_1}{2\pi} \int_{-\infty - i\epsilon_2}^{\infty - i\epsilon_2} \frac{d\omega_2}{2\pi} \int_{-\infty + i\epsilon_2}^{\infty + i\epsilon_2} \frac{d\omega_3}{2\pi} \int_{-\infty + i\epsilon_1}^{\infty + i\epsilon_1} \frac{d\omega_4}{2\pi}, \tag{83}$$

where $\epsilon_1$ and $\epsilon_2$ are infinitesimally small positive numbers. We provide the dependence of the integrand $\widetilde{\mathcal{I}}[g]$ on the spectral functions for all the quartic couplings in tables 2 and 3.

---

[21]The domain $\mathcal{C}_4$ in frequency space is determined by the time domain over which the integrals in (33) and (129) are defined.

Table 2: Couplings with even parity under time-reversal

| $g$ | $\widetilde{\mathcal{I}}[g]$ |
|---|---|
| $\overline{\lambda}_4 - \frac{1}{2}\widetilde{\kappa}_4$ | $\frac{3}{\omega_1\omega_4(\omega_3+\omega_4)}\big(-\rho[1234]+\rho[4321]\big)$ |
| $\overline{\zeta}_\mu^{(2)} - \frac{1}{24}\overline{\lambda}_4$ | $\frac{1}{\omega_1\omega_4(\omega_3+\omega_4)}\big(\rho\langle 1234\rangle - \rho\langle 4321\rangle\big)$ |
| $\zeta_N^{(4)} + 3\widetilde{\varrho}_4$ | $-\frac{3i}{2\omega_1\omega_4(\omega_3+\omega_4)}\big(\rho[12_+3_+4_+]+\rho[43_+2_+1_+]\big)$ |
| $\overline{\zeta}_3 - \frac{1}{3}\zeta_N^{(4)}$ | $\frac{4i}{\omega_1\omega_4(\omega_3+\omega_4)}\big(\rho\langle 1234\rangle + \rho\langle 4321\rangle\big)$ |
| $\kappa_4$ | $-\frac{1}{2\omega_1\omega_4(\omega_3+\omega_4)}\Big[\big(\rho[1234]-\rho[4321]\big)-\frac{1}{2}\big(\rho[1423]-\rho[4132]+\rho[2314]-\rho[3241]\big)\Big]$ |
| $\varrho_4$ | $\frac{i}{2\omega_1\omega_4(\omega_3+\omega_4)}\big(-\rho[12_+3_+4_+]-\rho[4_+3_+2_+1_+]+\rho[14_+2_+3_+]+\rho[41_+3_+2_+]+\rho[2314_+]+\rho[3241_+]\big)$ |
| $\overline{\lambda}_{4\gamma} + 2\widetilde{\kappa}_{4\gamma}^I$ | $\frac{i}{\omega_1^2\omega_4^2(\omega_3+\omega_4)^2}\Big[\big(\omega_1(\omega_2+\omega_1)-\omega_4(3\omega_2+5\omega_1)\big)\rho[1234]$ $-\big(\omega_4(\omega_3+\omega_4)-\omega_1(3\omega_3+5\omega_4)\big)\rho[4321]\Big]$ |
| $\overline{\zeta}_{3\gamma} - \frac{1}{2}\widetilde{\varrho}_{4\gamma}^I$ | $\frac{1}{4\omega_1^2\omega_4^2(\omega_3+\omega_4)^2}\Big[\big((\omega_3+\omega_4)(\omega_4-\omega_1)-2\omega_1\omega_4\big)\big(\rho[12_+34]+\rho[43_+21]\big)$ $+(\omega_3+\omega_4)(\omega_4-\omega_1)\big(\rho[123_+4]+\rho[432_+1]\big)$ $+(\omega_3+\omega_4)(\omega_4+\omega_1)\big(\rho[1234_+]-\rho[4321_+]\big)\Big]$ |
| $\kappa_{4\gamma}^I$ | $\frac{i}{4\omega_1^2\omega_4^2(\omega_3+\omega_4)^2}\Big[\big((\omega_3+\omega_4)(\omega_1-\omega_4)+2\omega_1\omega_4\big)\big(\rho[12_+][34_+]-\rho[43_+][21_+]\big)$ $+(\omega_3+\omega_4)(\omega_1-\omega_4)\big(\rho[13_+][24_+]-\rho[42_+][31_+]\big)$ $-(\omega_3+\omega_4)(\omega_4+\omega_1)\big(\rho[14_+][23_+]-\rho[32_+][41_+]\big)\Big]$ |
| $\kappa_{4\gamma}^{II}$ | $\frac{i}{4\omega_1^2\omega_4^2(\omega_3+\omega_4)^2}\Big[\big((\omega_3+\omega_4)(\omega_1-\omega_4)+2\omega_1\omega_4\big)\big(\rho[12][34]-\rho[43][21]\big)$ $+(\omega_3+\omega_4)(\omega_1-\omega_4)\big(\rho[13][24]-\rho[42][31]\big)$ $-(\omega_3+\omega_4)(\omega_4+\omega_1)\big(\rho[14][23]-\rho[32][41]\big)\Big]$ |
| $\kappa_{4\gamma}^{III}$ | $\frac{i}{8\omega_1^2\omega_4^2(\omega_3+\omega_4)^2}\Big[\big((\omega_3+\omega_4)(\omega_1-\omega_4)+2\omega_1\omega_4\big)\big(\rho[4321]-\rho[1234]+\rho[2314]-\rho[3241]+\rho[2413]-\rho[3142]\big)$ $+(\omega_3+\omega_4)(\omega_1-\omega_4)\big(\rho[4231]-\rho[1324]+\rho[2341]-\rho[3214]+\rho[3412]-\rho[2143]\big)\Big]$ |
| $\varrho_{4\gamma}^I$ | $\frac{1}{\omega_1^2\omega_4^2(\omega_3+\omega_4)^2}\Big[-\big((\omega_3+\omega_4)(\omega_1-\omega_4)+2\omega_1\omega_4\big)\big(\rho[13][24]+\rho[42][31]+\rho[14][23]+\rho[32][41]\big)$ $-(\omega_3+\omega_4)(\omega_1-\omega_4)\big(\rho[12][34]+\rho[43][21]\big)\Big]$ |
| $\varrho_{4\gamma}^{II}$ | $\frac{1}{4\omega_1^2\omega_4^2(\omega_3+\omega_4)^2}\Big[\big((\omega_3+\omega_4)(\omega_1-\omega_4)+2\omega_1\omega_4\big)\big(\rho[12_+][34]+\rho[43_+][21]+\rho[12][34_+]+\rho[43][21_+]\big)$ $+(\omega_3+\omega_4)(\omega_1-\omega_4)\big(\rho[13_+][24]+\rho[42_+][31]+\rho[13][24_+]+\rho[42][31_+]\big)$ $-(\omega_3+\omega_4)(\omega_1+\omega_4)\big(\rho[14_+][23]+\rho[32_+][41]+\rho[14][23_+]+\rho[32][41_+]\big)\Big]$ |

Table 3: Couplings with odd parity under time-reversal

| $g$ | $\widetilde{\mathcal{I}}[g]$ |
|---|---|
| $\widetilde{\kappa}_4$ | $-\frac{6}{\omega_1\omega_4(\omega_3+\omega_4)}\big(\rho[1234]+\rho[4321]\big)$ |
| $\widetilde{\varrho}_4$ | $-\frac{i}{2\omega_1\omega_4(\omega_3+\omega_4)}\Big[\big(\rho[12_+34]+\rho[123_+4]+\rho[1234_+]\big)-\big(\rho[43_+21]+\rho[432_+1]+\rho[4321_+]\big)\Big]$ |
| $\overline{\lambda}_{4\gamma} - 24\zeta_\gamma^{(2)}$ $-8(\kappa_{4\gamma}^I - \kappa_{4\gamma}^{II})$ | $-\frac{2i}{\omega_1^2\omega_4^2(\omega_3+\omega_4)^2}\Big[\big((\omega_3+\omega_4)(\omega_1-\omega_4)+2\omega_1\omega_4\big)\big(\rho[12][34_+]+\rho[43][21_+]-\rho[12_+][34]-\rho[43_+][21]\big)$ $+(\omega_3+\omega_4)(\omega_1-\omega_4)\big(\rho[13][24_+]+\rho[42][31_+]-\rho[13_+][24]-\rho[42_+][31]\big)$ $-(\omega_3+\omega_4)(\omega_1+\omega_4)\big(\rho[14][23_+]+\rho[32][41_+]-\rho[14_+][23]-\rho[32_+][41]\big)\Big]$ |
| $\widetilde{\kappa}_{4\gamma}^I$ | $-\frac{i}{2\omega_1^2\omega_4^2(\omega_3+\omega_4)^2}\Big[\big(\omega_1(\omega_2+\omega_1)-\omega_4(3\omega_2+5\omega_1)\big)\rho[1234]$ $+\big(\omega_4(\omega_3+\omega_4)-\omega_1(3\omega_3+5\omega_4)\big)\rho[4321]\Big]$ |
| $\widetilde{\kappa}_{4\gamma}^{II}$ | $\frac{i}{2\omega_1^2\omega_4^2(\omega_3+\omega_4)^2}\Big[(\omega_1-\omega_4)\big(\rho[23][14]-\rho[14][23]\big)+(\omega_1+\omega_4)\big(\rho[12][34]-\rho[43][21]+\rho[13][24]-\rho[42][31]\big)\Big]$ |
| $\widetilde{\varrho}_{4\gamma}^I$ | $-\frac{1}{2\omega_1^2\omega_4^2(\omega_3+\omega_4)^2}\Big[\big((\omega_1-\omega_4)(\omega_3+\omega_4)+2\omega_1\omega_4\big)\big(\rho[12_+34]-\rho[43_+21]\big)$ $+(\omega_1-\omega_4)(\omega_3+\omega_4)\big(\rho[123_+4]-\rho[432_+1]\big)$ $-(\omega_1+\omega_4)(\omega_3+\omega_4)\big(\rho[1234_+]+\rho[4321_+]\big)\Big]$ |
| $\widetilde{\varrho}_{4\gamma}^{II}$ | $\frac{1}{2\omega_1^2\omega_4^2(\omega_3+\omega_4)^2}\Big[-\big((\omega_1-\omega_4)(\omega_3+\omega_4)+\omega_1\omega_4\big)\big(\rho[14][23_+]+\rho[32_+][41]\big)+\omega_1\omega_4\big(\rho[14_+][23]+\rho[32][41_+]\big)$ $+\omega_1(\omega_3+2\omega_4)\big(\rho[13_+][24]+\rho[42][31_+]\big)-\omega_4(\omega_2+2\omega_1)\big(\rho[42_+][31]+\rho[13][24_+]\big)$ $+\omega_1(\omega_3+\omega_4)\big(\rho[12_+][34]+\rho[43][21_+]\big)-\omega_4(\omega_2+\omega_1)\big(\rho[43_+][21]+\rho[12][34_+]\big)\Big]$ |

Now, since all the couplings in tables 2 and 3 are real [22], one can obtain alternative expres-

---

[22]To see the reality of the OTO couplings, one can substitute relations like (34) in the expressions of these

sions for them by taking complex conjugates of the corresponding integrals. Such a complex conjugation results in the following transformations in the integral:

1. All explicit factors of $i$ go to $(-i)$.

2. In all factors involving the frequencies explicitly, $\omega_i$ goes to $\omega_i^*$ for $i \in \{1, 2, 3, 4\}$.

3. The spectral functions go to their complex conjugates.

Now, using the relation (72) which is based on the microscopic reversibility in the bath, one can obtain the following relations between the spectral functions:

$$
\begin{aligned}
(\rho\langle 1234\rangle)^* &= \rho\langle 1^*2^*3^*4^*\rangle, \\
(\rho[1234])^* &= \rho[1^*2^*3^*4^*], \\
(\rho[12][34])^* &= \rho[1^*2^*][3^*4^*], \text{etc.}
\end{aligned}
\tag{84}
$$

On the right hand sides of these equations, $i^*$ stands for $\omega_i^*$.

Hence, the net effect of the complex conjugation of the couplings is the replacement of all $\omega_i$'s by $\omega_i^*$'s and all explicit factors of $i$ by $(-i)$ in the corresponding frequency integrals. The complex conjugation of the frequencies takes the domain of integration to $\mathcal{C}_4^*$, the integral over which is defined as

$$
\int_{\mathcal{C}_4^*} \equiv \int_{-\infty+i\epsilon_1}^{\infty+i\epsilon_1} \frac{d\omega_1^*}{2\pi} \int_{-\infty+i\epsilon_2}^{\infty+i\epsilon_2} \frac{d\omega_2^*}{2\pi} \int_{-\infty-i\epsilon_2}^{\infty-i\epsilon_2} \frac{d\omega_3^*}{2\pi} \int_{-\infty-i\epsilon_1}^{\infty-i\epsilon_1} \frac{d\omega_4^*}{2\pi}.
\tag{85}
$$

Now, notice that the domain $\mathcal{C}_4^*$ gets mapped exactly back to $\mathcal{C}_4$ under the following relabeling of the frequencies:

$$
\omega_1^* \to \omega_4, \ \omega_2^* \to \omega_3, \ \omega_3^* \to \omega_2, \ \omega_4^* \to \omega_1.
\tag{86}
$$

Therefore, to summarise, the alternative expressions for the couplings can be obtained by integrating over the same domain $\mathcal{C}_4$, but performing the following transformations on the integrands in the original expressions:

1. $i \to -i$ for all explicit factors of $i$,

2. $\omega_1 \leftrightarrow \omega_4, \ \omega_2 \leftrightarrow \omega_3$.

We define the action of time-reversal on the couplings to be the above transformations on the integrands in their expressions. As we saw, the equality between the couplings and their time-reversed counterparts crucially relies on the relations given in (84) which are based on the microscopic reversibility in the bath's dynamics.

Now, one can compare the expressions of the couplings given in tables 2 and 3 with the expressions of their time-reversed counterparts. While making these comparisons, it is important to bear in mind that the spectral functions include a delta function corresponding to energy conservation in the unperturbed dynamics of the bath. This allows one to impose conditions such as

$$
(\omega_1 + \omega_2) = -(\omega_3 + \omega_4)
\tag{87}
$$

within the expressions of the integrands. Taking this into account one can easily see, that the expressions for the couplings in table 2 remain unchanged under time-reversal. On the other hand, the expressions for the couplings in table 3 pick up a minus sign under time-reversal. Thus, all the couplings mentioned in table 2 have even parity under time-reversal, whereas the couplings in table 3 have odd parity under the same. All the couplings with odd parity

---

couplings given in table 6.

must go to zero due to the microscopic reversibility in the bath's dynamics. This imposes the following constraints on the effective couplings:

$$\widetilde{\kappa}_4 = \widetilde{\varrho}_4 = \widetilde{\kappa}_{4\gamma}^I = \widetilde{\kappa}_{4\gamma}^{II} = \widetilde{\varrho}_{4\gamma}^I = \widetilde{\varrho}_{4\gamma}^{II} = 0, \tag{88}$$

$$\overline{\lambda}_{4\gamma} - 24\zeta_\gamma^{(2)} - 8(\kappa_{4\gamma}^I - \kappa_{4\gamma}^{II}) = 0. \tag{89}$$

Since these constraints are based on the microscopic reversibility in the bath, one can think of them as generalisations of the Onsager reciprocal relations [46, 47] . These generalised Onsager relations are extensions of similar relations obtained between the cubic effective couplings in [30].

As we mentioned earlier, apart from the microscopic reversibility in the bath's dynamics, there is another source of relations between the effective couplings of the particle, viz. the thermality of the bath. In the next subsection, we will discuss how the KMS relations [7–9] between the thermal correlators of the bath lead to constraints on the effective couplings. We will see that these constraints, when combined with the relation given in (89), give rise to a generalised fluctuation-dissipation relation between the thermal jitter in the particle's damping and the non-Gaussianity in the distribution of the noise.

## 4.4 Generalised fluctuation-dissipation relation

In section 3.5, we saw that if we ignore the quartic couplings in the Schwinger-Keldysh effective theory, the corresponding stochastic dual reduces to a linear Langevin dynamics (see (38) and (39)). Under this dynamics, the particle experiences a damping as well as a random force drawn from a Gaussian distribution. These two forces are related to each other as both of them arise from the interaction with the bath. More precisely, in the high temperature limit of the bath, the relation between these forces is given by

$$\langle f^2 \rangle = \frac{2}{\beta}\gamma, \tag{90}$$

where $\langle f^2 \rangle$ is the strength of the thermal noise experienced by the particle and $\gamma$ is its damping coefficient. This relation is commonly known as the 'fluctuation-dissipation relation'.

The ideas leading to the discovery of this relation emerged at the beginning of the 20th century with the works of Einstein, Smoluchowski and Sutherland. A similar relation between the thermal fluctuation and the resistance in an electric conductor was experimentally observed by Johnson [23] , and then theoretically derived by Nyquist [24]. Later, a proof of such relations for more general systems was provided by Callen and Welton [25], which was further extended by Stratonovich [26, 56].

As discussed in [7, 49], the fluctuation-dissipation relation is a consequence of certain relations between the 2-point thermal correlators of the bath, which are now commonly known as the Kubo-Martin-Schwinger (KMS) relations [7, 8]. These relations were studied for higher point thermal correlators in [9]. There, it was observed that such KMS relations can connect the thermal OTOCs of a bath to its Schwinger-Keldysh correlators. This indicated the need for including the effects of the bath's OTOCs while exploring the possibility of finding generalisations of the fluctuation-dissipation relation.

To include the effects of the OTOCs of the bath, an OTO effective theory of the particle was developed up to cubic terms in [45]. The cubic couplings in this effective theory receive contributions from the 3-point correlators of the bath. In [30], it was shown that the KMS relations between these 3-point thermal correlators lead to a relation between two couplings

in the cubic OTO effective dynamics of the particle. When combined with the constraints imposed by microscopic reversibility in the bath, this relation leads to a generalisation of the fluctuation-dissipation relation. From the perspective of the dual stochastic theory, this generalised fluctuation-dissipation relation connects the thermal jitter in the particle's damping and the non-Gaussianity in the noise.

We will see that a similar generalised fluctuation-dissipation relation holds between the quartic effective couplings as well. Before discussing this relation, we will first review the KMS relations between thermal correlators of the bath and then show how they lead to the fluctuation-dissipation relation given in equation (90).

**Kubo-Martin-Schwinger relations:**
The KMS relations [7–9] connect all thermal correlators of the bath which can be obtained from each other by cyclic permutations of insertions. For example, consider the following n-point correlator

$$\langle O(t_1)O(t_2)\cdots O(t_n)\rangle = \mathrm{Tr}\Big[\frac{e^{-\beta H_B}}{Z_B}O(t_1)O(t_2)\cdots O(t_n)\Big].$$
(91)

In the above expression, if we bring the insertion $O(t_n)$ from the right-most position to the left-most position across the thermal density matrix, the argument of the insertion picks up an extra term $(-i\beta)$ i.e.

$$\langle O(t_1)O(t_2)\cdots O(t_n)\rangle = \langle O(t_n - i\beta)O(t_1)\cdots O(t_{n-1})\rangle.$$
(92)

In frequency space, these relations lead to the following kind of relations between the spectral functions:

$$\rho\langle 12\cdots n\rangle = e^{-\beta\omega_n}\rho\langle n\, 1\cdots(n-1)\rangle.$$
(93)

Here $\rho\langle 12\cdots n\rangle$ and $\rho\langle n\, 1\cdots(n-1)\rangle$ are defined in terms of the $n$-point cumulants of $O$ in the time domain as follows:

$$\int_{-\infty}^{\infty}\frac{d\omega_1}{2\pi}\cdots\int_{-\infty}^{\infty}\frac{d\omega_n}{2\pi}\rho\langle 12\cdots n\rangle e^{-i(\omega_1 t_1+\cdots+\omega_n t_n)} \equiv \lambda^n\langle O(t_1)O(t_2)\cdots O(t_n)\rangle_c,$$

$$\int_{-\infty}^{\infty}\frac{d\omega_1}{2\pi}\cdots\int_{-\infty}^{\infty}\frac{d\omega_n}{2\pi}\rho\langle n\, 1\cdots(n-1)\rangle e^{-i(\omega_1 t_1+\cdots+\omega_n t_n)} \equiv \lambda^n\langle O(t_n)O(t_1)\cdots O(t_{n-1})\rangle_c.$$
(94)

In general, there are $n!$ such spectral functions corresponding to all the $n$-point Wightman correlators of $O$. However, KMS relations like (93) reduce the number of independent n-point spectral functions to $(n-1)!$.

Let us now review how such KMS relations between the bath's correlators lead to the fluctuation-dissipation relation given in (90).

**Fluctuation-dissipation relation between quadratic couplings:**
The couplings in the quadratic terms of the effective action receive contributions from the 2-point cumulants of the operator $O$ as shown in (32). These quadratic couplings up to leading order in $\lambda$ can be re-expressed in terms of two spectral functions $\rho[12]$ and $\rho[12_+]$ which are defined as

$$\int_{-\infty}^{\infty}\frac{d\omega_1}{2\pi}\int_{-\infty}^{\infty}\frac{d\omega_2}{2\pi}\rho[12]e^{-i(\omega_1 t_1+\omega_2 t_2)} \equiv \lambda^2\langle[O(t_1),O(t_2)]\rangle_c,$$

$$\int_{-\infty}^{\infty}\frac{d\omega_1}{2\pi}\int_{-\infty}^{\infty}\frac{d\omega_2}{2\pi}\rho[12_+]e^{-i(\omega_1 t_1+\omega_2 t_2)} \equiv \lambda^2\langle\{O(t_1),O(t_2)\}\rangle_c.$$
(95)

We provide the expressions for these leading order forms of the quadratic couplings below:

$$Z_I = -\int_{\mathcal{C}_2} \frac{\rho[12_+]}{i\omega_1^3}, \ \Delta\bar{\mu}^2 = -\int_{\mathcal{C}_2} \frac{\rho[12]}{\omega_1}, \ \langle f^2 \rangle = \int_{\mathcal{C}_2} \frac{\rho[12_+]}{i\omega_1}, \ \gamma = \int_{\mathcal{C}_2} \frac{\rho[12]}{i\omega_1^2}. \quad (96)$$

Here, the integrals are performed over the following domain

$$\int_{\mathcal{C}_2} \equiv \int_{-\infty-i\epsilon}^{\infty-i\epsilon} \frac{d\omega_1}{2\pi} \int_{-\infty+i\epsilon}^{\infty+i\epsilon} \frac{d\omega_2}{2\pi}, \quad (97)$$

where $\epsilon$ is a small positive number.

Now, the KMS relations connect the two spectral functions as follows

$$\rho[12_+] = \coth\left(\frac{\beta\omega_1}{2}\right)\rho[12]. \quad (98)$$

Then, in the high temperature limit of the bath i.e the small $\beta$ limit, the above relation reduces to

$$\rho[12_+] = \left(\frac{2}{\beta\omega_1}\right)\rho[12], \quad (99)$$

where we take the leading order (in $\beta$) forms of the two spectral functions. Plugging this relation into the expressions of the couplings given in (96), we get

$$Z_I = -\frac{2}{\beta}\int_{\mathcal{C}_2} \frac{\rho[12]}{i\omega_1^4}, \ \Delta\bar{\mu}^2 = -\int_{\mathcal{C}_2} \frac{\rho[12]}{\omega_1}, \ \langle f^2 \rangle = \frac{2}{\beta}\int_{\mathcal{C}_2} \frac{\rho[12]}{i\omega_1^2}, \ \gamma = \int_{\mathcal{C}_2} \frac{\rho[12]}{i\omega_1^2}. \quad (100)$$

From these expressions, one can clearly see that at this high temperature limit,

$$\boxed{\langle f^2 \rangle = \frac{2}{\beta}\gamma,} \quad (101)$$

which is the fluctuation-dissipation relation that we mentioned earlier.

Let us now discuss how one can obtain a generalisation of this fluctuation-dissipation relation for the quartic couplings.

**Generalised fluctuation-dissipation relation between quartic couplings:**

As pointed out in [9], the KMS relations between the 4-point functions of the bath can connect OTOCs to Schwinger-Keldysh correlators. For example, consider the correlator

$$\langle O(t_1)O(t_3)O(0)O(t_2)\rangle \equiv \text{Tr}\left[\frac{e^{-\beta H_B}}{Z_B}O(t_1)O(t_3)O(0)O(t_2)\right], \quad (102)$$

where $t_1 > t_2 > t_3 > 0$. Now, notice that this correlator satisfies the following KMS relation:

$$\langle O(t_1)O(t_3)O(0)O(t_2) = \langle O(t_2 - i\beta)O(t_1)O(t_3)O(0)\rangle. \quad (103)$$

This KMS relation connects the correlator given in (102) to the following correlator by analytic continuation:

$$\langle O(t_2)O(t_1)O(t_3)O(0)\rangle \equiv \text{Tr}\left[\frac{e^{-\beta H_B}}{Z_B}O(t_2)O(t_1)O(t_3)O(0)\right]. \quad (104)$$

The correlator given in (102) is an OTOC which can be obtained by putting insertions on the 2-fold contour as shown in figure 6. On the other hand, the correlator given in (104) is a

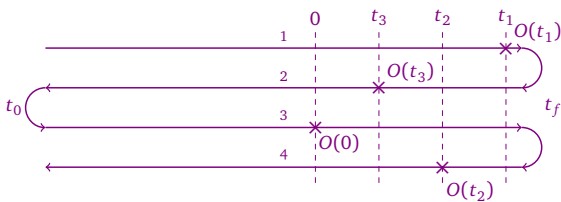

Figure 6: Contour-ordered correlator for $\langle O(t_1)O(t_3)O(0)O(t_2)\rangle$ where $t_1 > t_2 > t_3 > 0$

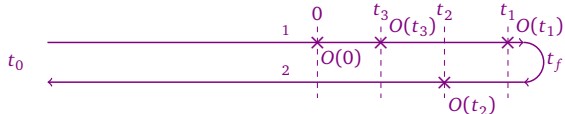

Figure 7: Contour-ordered correlator for $\langle O(t_2)O(t_1)O(t_3)O(0)\rangle$ where $t_1 > t_2 > t_3 > 0$

Schwinger-Keldysh correlator as demonstrated in figure 7.

We will see that KMS relations like (103) which connect OTO correlators of the bath to Schwinger-Keldysh correlators result in a relation between a quartic OTO coupling and a Schwinger-Keldysh coupling of the particle. To derive this relation, we need to go back to the expressions of the particle's effective couplings in terms of the bath's spectral functions given in tables 2 and 3. As discussed earlier, the KMS relations reduce the number of independent 4-point spectral functions to $(4-1)! = 6$. We choose these 6 independent spectral functions to be the following[23]:

$$\rho[1234],\ \rho[4321],\ \rho[2314],\ \rho[3241],\ \rho[2143],\ \rho[3142]. \tag{105}$$

We provide the expressions of the quartic effective couplings in terms of these 6 spectral functions at the high temperature limit of the bath in appendix D. Among these expressions, we will focus on the forms of two particular couplings here: $\kappa_{4\gamma}^I$ and $\zeta_N^{(4)}$. From the expressions of these couplings given in tables 7 and 8, we can see that the corresponding integrands satisfy the following relation:

$$
\begin{aligned}
\widetilde{\mathcal{I}}[\kappa_{4\gamma}^I] &= \frac{\beta}{4}\widetilde{\mathcal{I}}[\zeta_N^{(4)}] \\
&= \frac{6i}{\beta^2\omega_1^2\omega_2\omega_3\omega_4^2(\omega_1+\omega_3)(\omega_1+\omega_4)(\omega_3+\omega_4)^2} \\
&\quad \Big[\omega_1(\omega_1+\omega_4)(\omega_4^2-\omega_2\omega_3)\rho[1234] + \omega_4(\omega_1+\omega_3)(\omega_1+\omega_4)(\omega_3+\omega_4)\rho[4321] \\
&\quad -\omega_1\omega_4(\omega_1+\omega_3)(\omega_1+\omega_4)\rho[2143] + \omega_1\omega_2(\omega_3^2-\omega_4^2)\rho[2314] \\
&\quad +\omega_1\omega_4(\omega_1+\omega_4)(\omega_3+\omega_4)\rho[3142] - \omega_4\omega_2(\omega_1+\omega_3)(\omega_3+\omega_4)\rho[3241]\Big],
\end{aligned}
\tag{106}
$$

where we take only the leading order (in $\beta$) forms for the spectral functions. From this relation between the integrands, we conclude that at leading order in $\lambda$ and $\beta$, these two couplings satisfy the following relation:

$$\boxed{\kappa_{4\gamma}^I = \frac{\beta}{4}\zeta_N^{(4)}.} \tag{107}$$

---

[23]See appendix D for a discussion on why these 6 spectral functions form a basis for all 4-point thermal correlators of $O$.

This is an example of a generalisation of the fluctuation-dissipation relation which connects an OTO coupling to a Schwinger-Keldysh coupling. From this analysis, one can see that this relation is purely a consequence of KMS relations between the bath's correlators and holds even when the bath's dynamics lacks microscopic reversibility.

In the presence of microscopic reversibility in the bath, there is an additional relation which involves the coupling $\kappa_{4\gamma}^I$. This is one of the generalised Onsager relations given in (89) which we quote here once more for convenience:

$$\overline{\lambda}_{4\gamma} - 24\zeta_\gamma^{(2)} - 8(\kappa_{4\gamma}^I - \kappa_{4\gamma}^{II}) = 0. \tag{108}$$

Now, referring to the expressions of the leading order forms of the couplings $\overline{\lambda}_{4\gamma}$ and $\kappa_{4\gamma}^{II}$ given in tables 7 and 8, we see that the corresponding integrands are as follows:

$$\widetilde{\mathcal{I}}[\overline{\lambda}_{4\gamma}] = \frac{2i}{\omega_1^2\omega_4^2(\omega_3+\omega_4)^2}\Big\{3(\omega_4-\omega_1)(\omega_3+\omega_4)+2\omega_1\omega_3\Big\}\rho[1234],$$

$$\begin{aligned}
\widetilde{\mathcal{I}}[\kappa_{4\gamma}^{II}] = \frac{i}{4\omega_1^2\omega_4^2(\omega_3+\omega_4)^2}\Big[&(\omega_1-\omega_4)(\omega_3+\omega_4)\big(\rho[1234]+\rho[2314]+\rho[3142]\big)\\
&+(\omega_1+\omega_4)(\omega_3+\omega_4)\big(\rho[2314]+\rho[3241]\big)\\
&+\big((\omega_1-\omega_4)(\omega_3+\omega_4)+2\omega_1\omega_4\big)\big(\rho[1234]+\rho[2143]\big)\Big].
\end{aligned} \tag{109}$$

Notice that these integrands are suppressed by a factor of $\beta^2$ compared to the integrand for $\kappa_{4\gamma}^I$ given in (106). Therefore, in the high temperature limit, we can ignore the couplings $\overline{\lambda}_{4\gamma}$ and $\kappa_{4\gamma}^{II}$ in (89) to get

$$\boxed{\kappa_{4\gamma}^I = -3\zeta_\gamma^{(2)}.} \tag{110}$$

Combining the equations (110) and (107), we get the following relation in the high temperature limit:

$$\boxed{\zeta_N^{(4)} = -\frac{12}{\beta}\zeta_\gamma^{(2)}.} \tag{111}$$

This is a generalised fluctuation-dissipation relation which connects the non-Gaussianity in the noise experienced by the particle to the thermal jitter in its damping coefficient. It is a combined effect of microscopic reversibility in the bath and its thermality. In the following subsection, we will compute the effective couplings in the $qXY$ model and verify the validity of this relation.

## 4.5 Values of the effective couplings

In this subsection, we will enumerate the values of the effective couplings of the Brownian particle in the $qXY$ model.

The forms of the quadratic couplings at leading order in $\lambda$ and $\beta$ are given in (100) in terms of the 2-point function $\rho[12]$. Similar forms for the quartic couplings are given in tables 7 and 8 in terms of the 4-point spectral functions enumerated in (105).

We provide the leading order (in $\beta$) forms of the 2-point spectral function $\rho[12]$ and the 4-point spectral function $\rho[1234]$ for the $qXY$ model in (112) and (113) respectively.

$$\rho[12] = 2\pi\delta(\omega_1+\omega_2)\frac{\Gamma_2}{\beta}\frac{4\omega_1\Omega^3}{\omega_1^2+4\Omega^2}. \tag{112}$$

$$\rho[1234] = 2\pi\delta(\omega_1 + \omega_2 + \omega_3 + \omega_4)\left(16i\frac{\Gamma_4\Omega^7}{\beta}\right)$$

$$\left[\left\{\frac{\left(\Omega - i\omega_3\right)\left(\Omega - i(\omega_1 + \omega_3)\right)}{\omega_3\left(\omega_1 + \omega_3\right)\left(2\Omega - i\omega_3\right)\left(2\Omega + i\omega_4\right)\left(2\Omega - i(\omega_1 + \omega_3)\right)}\right.\right.$$

$$-\frac{\left(\Omega + i\omega_2\right)\left(\Omega + i(\omega_1 + \omega_2)\right)}{\omega_2\left(\omega_1 + \omega_2\right)\left(2\Omega + i\omega_2\right)\left(2\Omega - i\omega_4\right)\left(2\Omega + i(\omega_1 + \omega_2)\right)}$$

$$+\frac{\omega_4\Omega\left(\Omega - i\omega_1\right)}{\omega_1\omega_3\left(\omega_1 + \omega_2\right)\left(2\Omega - i\omega_1\right)\left(2\Omega + i\omega_3\right)\left(2\Omega - i(\omega_1 + \omega_2)\right)}$$

$$\left.-\frac{\omega_4\Omega\left(\Omega + i\omega_1\right)}{\omega_1\omega_2\left(\omega_1 + \omega_3\right)\left(2\Omega + i\omega_1\right)\left(2\Omega - i\omega_2\right)\left(2\Omega + i(\omega_1 + \omega_3)\right)}\right\}$$

$$\left.-\left\{(\omega_1 \leftrightarrow \omega_2)\right\}\right]. \tag{113}$$

The other five 4-point spectral functions can be obtained from (113) by appropriate permutations of the frequencies.

Substituting these spectral functions in the integrals for the couplings, one can calculate the values of these couplings in the high temperature limit. We provide the values of the quadratic couplings in (114), the Schwinger-Keldysh quartic couplings in (115), and the OTO quartic couplings in (116) and (117).

**Quadratic couplings:**

$$\boxed{Z_I = \frac{\Gamma_2}{4\beta^2\Omega} \, , \, \Delta\bar{\mu}^2 = -\frac{\Gamma_2\Omega^2}{\beta} \, , \, \langle f^2 \rangle = \frac{\Gamma_2\Omega}{\beta^2} \, , \, \gamma = \frac{\Gamma_2\Omega}{2\beta} \, .} \tag{114}$$

**Quartic couplings:**

A) **Schwinger-Keldysh couplings:**

$$\boxed{\begin{aligned} &\zeta_N^{(4)} = -\frac{15\Gamma_4\Omega}{\beta^4}, \, \bar{\lambda}_4 = -\frac{6\Gamma_4\Omega^4}{\beta}, \, \bar{\zeta}_3 = -\frac{6\Gamma_4\Omega^3}{\beta^2}, \, \zeta_\mu^{(2)} = -\frac{5\Gamma_4\Omega^2}{2\beta^3}, \\ &\bar{\lambda}_{4\gamma} = -\frac{6\Gamma_4\Omega^3}{\beta}, \, \bar{\zeta}_{3\gamma} = -\frac{15\Gamma_4\Omega^2}{4\beta^2}, \, \zeta_\gamma^{(2)} = \frac{5\Gamma_4\Omega}{4\beta^3}. \end{aligned}} \tag{115}$$

B) **OTO couplings:**

$$\boxed{\widetilde{\kappa}_4 = \widetilde{\varrho}_4 = \widetilde{\kappa}_{4\gamma}^I = \widetilde{\kappa}_{4\gamma}^{II} = \widetilde{\varrho}_{4\gamma}^I = \widetilde{\varrho}_{4\gamma}^{II} = 0.} \tag{116}$$

$$\boxed{\begin{aligned} &\kappa_4 = -\frac{3\Gamma_4\Omega^4}{2\beta}, \, \varrho_4 = \frac{7\Gamma_4\Omega^3}{2\beta^2}, \, \kappa_{4\gamma}^I = -\frac{15\Gamma_4\Omega}{4\beta^3}, \, \kappa_{4\gamma}^{II} = \frac{5\Gamma_4\Omega^3}{8\beta}, \\ &\kappa_{4\gamma}^{III} = -\frac{19\Gamma_4\Omega^3}{16\beta}, \, \varrho_{4\gamma}^I = -\frac{15\Gamma_4\Omega^2}{4\beta^2}, \, \varrho_{4\gamma}^{II} = \frac{5\Gamma_4\Omega^2}{2\beta^2}. \end{aligned}} \tag{117}$$

From the values of these couplings, one can easily verify the validity of the fluctuation dissipation relation (101), the generalised fluctuation dissipation relation (111), and the generalised Onsager relations (88) and (110) in the $qXY$ model.

# 5  Conclusion and discussion

In this paper, we have developed the quartic effective dynamics of a Brownian particle weakly interacting with a thermal bath. To illustrate the features of this effective dynamics, we have introduced a simple toy model (the $qXY$ model described in section 2) where the bath comprises of two sets of harmonic oscillators coupled to the particle through cubic interactions.

For this model, we have identified a Markovian regime, where the particle's effective dynamics is approximately local in time. Working in this regime, we have constructed a quartic effective action of the particle in the Schwinger-Keldysh (SK) formalism. Using the techniques developed in [52–54], we have demonstrated a duality between this quantum effective theory and a classical stochastic dynamics governed by a non-linear Langevin equation.

The SK effective theory and the dual non-linear Langevin dynamics suffer from the limitation that they provide no information about the 4-point out of time order correlators (OTOCs) of the particle. To transcend this limitation, we have extended the SK effective action to an out of time ordered effective action defined on a generalised Schwinger-Keldysh contour (see figure 3). In this extended framework, we have determined the additional quartic couplings which encode the effects of the bath's 4-point OTOCs on the particle's dynamics. We have worked out the dependence of these OTO couplings (as well as the SK effective couplings) on the correlators of the bath up to leading order in the particle-bath interaction.

The relations between the particle's effective couplings and the bath's correlators provide a way to analyse the constraints imposed on the effective dynamics due to thermality and microscopic reversibility of the bath. These constraints manifest in the form of certain relations between the quartic couplings which can be interpreted as OTO generalisations of the well-known Onsager reciprocal relations and fluctuation-dissipation relation (FDR). By combining these relations, we have obtained a generalised FDR which connects two of the Schwinger-Keldysh effective couplings. In the dual stochastic dynamics, these two couplings correspond to a thermal jitter in the damping coefficient and a non-Gaussianity in the noise distribution. The generalised FDR between these two quartic couplings is an extension of a similar relation obtained for the cubic effective dynamics in [30].

The generalised FDRs and Onsager relations in both the cubic and the quartic effective theories of the particle suggest that such relations probably hold for even higher degree terms in the effective action when the bath's microscopic dynamics is reversible. It would be interesting to identify the general form of these relations.

Although the construction of the quartic effective theory in this paper is demonstrated with the $qXY$ model, the analysis mostly relies on the validity of the Markov approximation for the particle's dynamics. Hence, it may be employed to study the effective theory of the particle when it interacts with more complicated baths. For instance, the bath may even be a strongly coupled system [24] in which case a microscopic analysis of the particle's dynamics is very difficult. In such a scenario, the quartic effective theory of the particle would allow one to determine the particle's 4-point correlators (including its OTOCs) in terms of the effective couplings.

For the qXY model studied in this paper, the bath's 4-point cumulants decay exponentially when the time interval between any two insertions is increased. This allowed us to work in a

---

[24]Notice that we have assumed a weak coupling only between the particle and the bath. The couplings between the internal degrees of freedom of the bath may be strong.

Markovian regime by tuning the parameters in the model such that the particle's evolution is much slower than the decay of the bath's cumulants. However, as pointed out in [45], such an exponential damping of the bath's cumulants is not strictly necessary in all time regimes for obtaining a nearly local dynamics of the particle. In fact, the Markov approximation for the particle's dynamics may be valid even for a chaotic bath [25] [57–60] as long as the bath's OTO cumulants saturate to sufficiently small values much faster than the particle's evolution [45]. This opens up the possibility of probing the Lyapunov exponents [35] in such chaotic baths by measuring the OTO effective couplings of the particle [45] (See the relations between the particle's OTO effective couplings and the bath's OTOCs given in (129) and table 6).

The applicability of our effective theory framework to the scenario where the bath is chaotic and strongly coupled implies that it may be possible to construct a holographic dual description [61–69] of the particle's non-linear dynamics. The OTO extension of this non-linear dynamics may be useful in determining a holographic prescription for computing the particle's OTOCs [67, 70].

The non-linear Langevin equation that we discussed in section 3.5 has a structural similarity with the equations of motion of damped anharmonic oscillators like the Van der Pol oscillator [71] and the Duffing oscillator [72] [26]. Such oscillators, under periodic driving, are known to exhibit chaos in appropriate parameter regimes [72–74]. It would be interesting to see whether one can find a similar regime in the non-linear Langevin dynamics where the particle undergoes a chaotic motion.

It will be useful to formulate a Wilsonian counterpart of the out of time ordered 1-PI effective action developed in this paper. Such a Wilsonian effective theory can be extended to open quantum field theories [51, 75–82] which show up in the study of quantum cosmology and heavy ion physics. It will be interesting to determine the RG flow [51] of the OTO couplings in this Wilsonian framework to estimate their relative importance at different energy scales.

# Acknowledgements

We would like to thank Bijay Kumar Agarwalla, Ahana Chakraborty, Deepak Dhar, Arghya Das, Abhijit Gadde, G J Sreejith, Sachin Jain, Chandan Jana, Dileep Jatkar, Anupam Kundu, R. Loganayagam, Rajdeep Sensarma and Spenta Wadia for useful discussions. We are grateful for the support from International Centre for Theoretical Sciences (ICTS-TIFR), Bangalore. BC would like to thank the Tata Institute of Fundamental Research (TIFR), Mumbai, the Indian Institute of Science Education and Research (IISER), Pune and the Harish-Chandra Research Institute (HRI), Prayagraj for hospitality towards the final stages of this work. BC acknowledges Infosys Program for providing travel support to various conferences. SC would like to thank the Kavli Institute of Theoretical Physics (KITP), UC Santa Barbara and The Abdus Salam International Centre for Theoretical Physics (ICTP), Trieste for hospitality during the course of this work. We acknowledge our debt to the people of India for their steady and generous support to research in the basic sciences.

**Funding information:**   This research was supported in part by the National Science Foundation under Grant No. NSF PHY-1748958.

---

[25]The OTO cumulants in such chaotic baths show an exponentially fast fall-off initially (in the Lyapunov regime) before saturating to some constant values.

[26]The major difference is the presence of a thermal noise in the Langevin dynamics.

# A Cumulants of the bath operator that couples to the particle

In this appendix, we provide the forms of the 2-point and 4-point cumulants of the bath operator $\lambda O(t) \equiv \lambda \sum_{i,j} g_{xy,ij} X^{(i)}(t) Y^{(j)}(t)$ that couples to the particle. These cumulants are calculated in the high temperature limit where

$$\beta \Omega \ll 1. \tag{118}$$

We will see that, in this limit, the cumulants decay exponentially when the separations between insertions are increased. To show the form of this decay we follow the notational conventions given below:

- The interval between two time instants $t_i$ and $t_j$ is expressed as

$$t_{ij} \equiv t_i - t_j. \tag{119}$$

- The cumulant of any Wightman correlator $\langle O(t_{i_1}) O(t_{i_2}) \cdots O(t_{i_n}) \rangle$ is expressed as

$$\langle i_1 i_2 \cdots i_n \rangle \equiv \langle O(t_{i_1}) O(t_{i_2}) \cdots O(t_{i_n}) \rangle_c. \tag{120}$$

Keeping these notational conventions in mind, let us now discuss the decaying behaviour of the cumulants.

## A.1 2-point cumulants

From the form of the operator $O(t)$ given in (3), we can see that the 2 point cumulants receive contributions from the Feynman diagram shown in figure 8. Here the blue and red lines

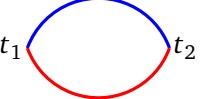

Figure 8: Feynman diagram contributing to the 2-point cumulants

represent $X^{(i)}$-$X^{(i)}$ and $Y^{(j)}$-$Y^{(j)}$ propagators respectively. To get the 2-point cumulants, one has to sum over all the oscillator frequencies in the bath. In the continuum limit, these sums reduce to integrals with the appropriate distribution of the couplings given in (8). Performing these integrals over the oscillator frequencies, we find that the cumulants decay exponentially when the time interval between the insertions is increased. In table 4, we provide the forms of the slowest decaying modes in these cumulants. The decay rates of these modes are of the order of $\Omega$. All the other modes which we have not mentioned in table 4 decay at much faster rates which are of the order of $\beta^{-1}$.

Table 4: Decay of 2-point cumulants for $t_1 > t_2$

| Cumulant | Slowest decaying mode |
|---|---|
| $\lambda^2 \langle 12 \rangle$ | $-\frac{\Gamma_2 \Omega^4}{(e^{i\beta\Omega}-1)^2} e^{-2\Omega t_{12}}$ |
| $\lambda^2 \langle 21 \rangle$ | $-\frac{\Gamma_2 \Omega^4}{(-1+e^{-i\beta\Omega})^2} e^{-2\Omega t_{12}}$ |

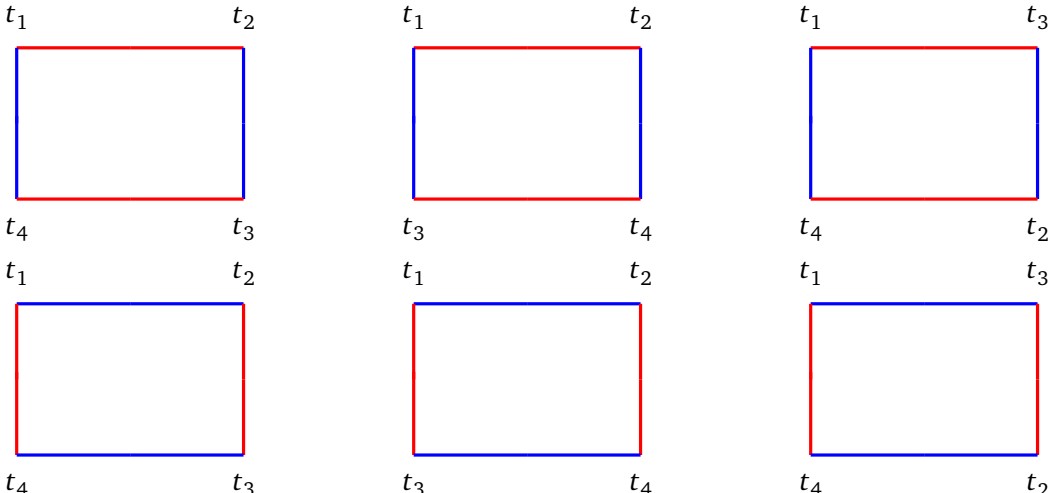

Figure 9: Feynman diagrams contributing to the 4-point cumulants

## A.2 4-point cumulants

The Feynman diagrams that contribute to the 4-point cumulants of $O(t)$ are given in figure 9. Again, integrating over all the oscillator frequencies with the corresponding couplings following the distribution given in (9), one can obtain the forms of these cumulants. As in case of the 2-point cumulants, these 4-point cumulants also decay exponentially when the separation between any two insertions is increased. We provide the forms of the slowest decaying modes of these cumulants in table 5. By looking at these forms, one can see that the decay rates of these modes are of the order of $\Omega$ as well.

## B Argument for the validity of the Markovian limit

In this appendix, we provide an argument for the validity of the Markov approximation for the parameter regime given in (20). This argument will mainly involve dimensional analysis.

In the units where $\hbar$ and the renormalised mass of the particle are unity, the dimension of each effective coupling can be expressed as some power of time. We provide these dimensions below.

**Dimensions of quadratic couplings:**

$$[Z_I] = T^0, \ [\gamma] = T^{-1}, \ [\bar{\mu}^2] = [\langle f^2 \rangle] = T^{-2} . \tag{121}$$

**Dimensions of SK quartic couplings:**

$$[\zeta_N^{(4)}] = [\bar{\lambda}_4] = [\bar{\zeta}_3] = [\zeta_\mu^{(2)}] = T^{-3}, \tag{122}$$
$$[\bar{\lambda}_{4\gamma}] = [\bar{\zeta}_{3\gamma}] = [\zeta_\gamma^{(2)}] = T^{-2}.$$

**Dimensions of OTO quartic couplings:**

$$[\kappa_4] = [\tilde{\kappa}_4] = [\varrho_4] = [\tilde{\varrho}_4] = T^{-3}, \tag{123}$$
$$[\kappa_{4\gamma}^I] = [\kappa_{4\gamma}^{II}] = [\kappa_{4\gamma}^{III}] = [\tilde{\kappa}_{4\gamma}^I] = [\tilde{\kappa}_{4\gamma}^{II}] = [\varrho_{4\gamma}^I] = [\varrho_{4\gamma}^{II}] = [\tilde{\varrho}_{4\gamma}^I] = [\tilde{\varrho}_{4\gamma}^{II}] = T^{-2}.$$

Table 5: Decay of 4-point cumulants for $t_1 > t_2 > t_3 > t_4$

| Cumulant | Slowest decaying mode |
|---|---|
| $\lambda^4 \langle 1234 \rangle$ | $\frac{2\Gamma_4 \Omega^8}{(e^{i\beta\Omega}-1)^4} e^{-2t_{14}\Omega} \left( e^{-2t_{23}\Omega} + 2 \right)$ |
| $\lambda^4 \langle 1243 \rangle$ | $\frac{2\Gamma_4 \Omega^8}{(e^{i\beta\Omega}-1)^4} e^{-2t_{14}\Omega} \left( e^{-2t_{23}\Omega} + 2e^{i\beta\Omega} \right)$ |
| $\lambda^4 \langle 1324 \rangle$ | $\frac{2\Gamma_4 \Omega^8}{(e^{i\beta\Omega}-1)^4} e^{-2t_{14}\Omega} \left( e^{-2t_{23}\Omega+i\beta\Omega} + e^{i\beta\Omega} + 1 \right)$ |
| $\lambda^4 \langle 1342 \rangle$ | $\frac{2\Gamma_4 \Omega^8}{(e^{i\beta\Omega}-1)^4} e^{-2t_{14}\Omega+2i\beta\Omega} \left( e^{-2t_{23}\Omega} + 2e^{-i\beta\Omega} \right)$ |
| $\lambda^4 \langle 1423 \rangle$ | $\frac{2\Gamma_4 \Omega^8}{(e^{i\beta\Omega}-1)^4} e^{-2t_{14}\Omega+i\beta\Omega} \left( 1 + e^{-2t_{23}\Omega} + e^{i\beta\Omega} \right)$ |
| $\lambda^4 \langle 1432 \rangle$ | $\frac{2\Gamma_4 \Omega^8}{(e^{i\beta\Omega}-1)^4} e^{-2t_{14}\Omega+2i\beta\Omega} \left( 2 + e^{-2t_{23}\Omega} \right)$ |
| $\lambda^4 \langle 2134 \rangle$ | $\frac{2\Gamma_4 \Omega^8}{(e^{i\beta\Omega}-1)^4} e^{-2t_{14}\Omega} \left( e^{-2t_{23}\Omega} + 2e^{i\beta\Omega} \right)$ |
| $\lambda^4 \langle 2143 \rangle$ | $\frac{2\Gamma_4 \Omega^8}{(e^{i\beta\Omega}-1)^4} e^{-2t_{14}\Omega} \left( e^{-2t_{23}\Omega} + 2e^{2i\beta\Omega} \right)$ |
| $\lambda^4 \langle 2314 \rangle$ | $\frac{2\Gamma_4 \Omega^8}{(e^{i\beta\Omega}-1)^4} e^{-2t_{14}\Omega+i\beta\Omega} \left( 1 + e^{-2t_{23}\Omega} + e^{i\beta\Omega} \right)$ |
| $\lambda^4 \langle 2341 \rangle$ | $\frac{2\Gamma_4 \Omega^8}{(e^{i\beta\Omega}-1)^4} e^{-2t_{14}\Omega+2i\beta\Omega} \left( e^{-2t_{23}\Omega} + 2 \right)$ |
| $\lambda^4 \langle 2413 \rangle$ | $\frac{2\Gamma_4 \Omega^8}{(e^{i\beta\Omega}-1)^4} e^{-2t_{14}\Omega+i\beta\Omega} \left( e^{-2t_{23}\Omega} + e^{i\beta\Omega} + e^{2i\beta\Omega} \right)$ |
| $\lambda^4 \langle 2431 \rangle$ | $\frac{2\Gamma_4 \Omega^8}{(e^{i\beta\Omega}-1)^4} e^{-2t_{14}\Omega+2i\beta\Omega} \left( e^{-2t_{23}\Omega} + 2e^{i\beta\Omega} \right)$ |
| $\lambda^4 \langle 3124 \rangle$ | $\frac{2\Gamma_4 \Omega^8}{(e^{i\beta\Omega}-1)^4} e^{-2t_{14}\Omega+2i\beta\Omega} \left( e^{-2t_{23}\Omega} + 2e^{-i\beta\Omega} \right)$ |
| $\lambda^4 \langle 3142 \rangle$ | $\frac{2\Gamma_4 \Omega^8}{(e^{i\beta\Omega}-1)^4} e^{-2t_{14}\Omega+i\beta\Omega} \left( 1 + e^{i\beta\Omega} + e^{-2t_{23}\Omega+2i\beta\Omega} \right)$ |
| $\lambda^4 \langle 3214 \rangle$ | $\frac{2\Gamma_4 \Omega^8}{(e^{i\beta\Omega}-1)^4} e^{-2t_{14}\Omega+2i\beta\Omega} \left( 2 + e^{-2t_{23}\Omega} \right)$ |
| $\lambda^4 \langle 3241 \rangle$ | $\frac{2\Gamma_4 \Omega^8}{(e^{i\beta\Omega}-1)^4} e^{-2t_{14}\Omega+2i\beta\Omega} \left( 1 + e^{i\beta\Omega} + e^{-2t_{23}\Omega+i\beta\Omega} \right)$ |
| $\lambda^4 \langle 3412 \rangle$ | $\frac{2\Gamma_4 \Omega^8}{(e^{i\beta\Omega}-1)^4} e^{-2t_{14}\Omega+2i\beta\Omega} \left( 2 + e^{-2t_{23}\Omega+2i\beta\Omega} \right)$ |
| $\lambda^4 \langle 3421 \rangle$ | $\frac{2\Gamma_4 \Omega^8}{(e^{i\beta\Omega}-1)^4} e^{-2t_{14}\Omega+3i\beta\Omega} \left( 2 + e^{-2t_{23}\Omega+i\beta\Omega} \right)$ |
| $\lambda^4 \langle 4123 \rangle$ | $\frac{2\Gamma_4 \Omega^8}{(e^{i\beta\Omega}-1)^4} e^{-2t_{14}\Omega+2i\beta\Omega} \left( 2 + e^{-2t_{23}\Omega} \right)$ |
| $\lambda^4 \langle 4132 \rangle$ | $\frac{2\Gamma_4 \Omega^8}{(e^{i\beta\Omega}-1)^4} e^{-2t_{14}\Omega+2i\beta\Omega} \left( 1 + e^{i\beta\Omega} + e^{-2t_{23}\Omega+i\beta\Omega} \right)$ |
| $\lambda^4 \langle 4213 \rangle$ | $\frac{2\Gamma_4 \Omega^8}{(e^{i\beta\Omega}-1)^4} e^{-2t_{14}\Omega+2i\beta\Omega} \left( 2e^{i\beta\Omega} + e^{-2t_{23}\Omega} \right)$ |
| $\lambda^4 \langle 4231 \rangle$ | $\frac{2\Gamma_4 \Omega^8}{(e^{i\beta\Omega}-1)^4} e^{-2t_{14}\Omega+3i\beta\Omega} \left( 1 + e^{i\beta\Omega} + e^{-2t_{23}\Omega} \right)$ |
| $\lambda^4 \langle 4312 \rangle$ | $\frac{2\Gamma_4 \Omega^8}{(e^{i\beta\Omega}-1)^4} e^{-2t_{14}\Omega+3i\beta\Omega} \left( 2 + e^{-2t_{23}\Omega+i\beta\Omega} \right)$ |
| $\lambda^4 \langle 4321 \rangle$ | $\frac{2\Gamma_4 \Omega^8}{(e^{i\beta\Omega}-1)^4} e^{-2t_{14}\Omega+4i\beta\Omega} \left( 2 + e^{-2t_{23}\Omega} \right)$ |

Now, from the value of each of these couplings given in section 4.5, one can obtain a time-scale. For the Markov approximation to be valid, we need all these time-scales to be much larger than $\Omega^{-1}$. This fixes the following Markovian regime for the parameters:

$$\overline{\mu}_0 \ll \Omega, \ \Gamma_2 \ll \beta(\beta\Omega), \ \Gamma_4 \ll \beta^2(\beta\Omega). \tag{124}$$

Since, the values of the parameters in section 4.5 are computed in the higher temperature limit, we need to impose the following condition as well:

$$\beta\Omega \ll 1 . \tag{125}$$

In addition to the above hierarchy, we need the following condition for the validity of the perturbative analysis [27]:

$$\Gamma_4 \ll \left( \Gamma_2 \right)^2 . \tag{126}$$

Note that imposing this condition along with the high temperature limit in (125) and the inequality for $\Gamma_2$ in (124) automatically ensures that $\Gamma_4$ falls in domain given in (124).

Combining all these conditions, we get the Markovian regime mentioned in (20).

---

[27] In this perturbative analysis, we assume that the contributions of the quartic terms to the particle's dynamics are small compared to those of the quadratic terms.

## C  Relations between the OTO effective couplings and the bath's OTOCs

In section 4.1, we obtained the form of the non-local generalised influence phase of the particle by integrating out the bath's degrees of freedom on the 2-fold contour. When the bath's cumulants decay sufficiently fast compared to the time-scales involved in the particle's evolution, one can get an approximately local form for this generalised influence phase. This approximately local form is similar to its Schwinger-Keldysh counterpart (see (21) and (22)). The quadratic and quartic terms in this approximately local generalised influence phase are as follows:

$$
\begin{aligned}
W_{GSK}^{(2)} \approx i \sum_{i_1,i_2=1}^{4} \int_{t_0}^{t_f} dt_1 \Big[ \Big\{ \int_{t_0}^{t_1} dt_2 \, \langle \mathcal{T}_C O_{i_1}(t_1) O_{i_2}(t_2) \rangle_c \Big\} q_{i_1}(t_1) q_{i_2}(t_1 - \epsilon) \\
+ \Big\{ \int_{t_0}^{t_1} dt_2 \, \langle \mathcal{T}_C O_{i_1}(t_1) O_{i_2}(t_2) \rangle_c \, t_{21} \Big\} q_{i_1}(t_1) \dot{q}_{i_2}(t_1 - \epsilon) \\
+ \Big\{ \int_{t_0}^{t_1} dt_2 \, \langle \mathcal{T}_C O_{i_1}(t_1) O_{i_2}(t_2) \rangle_c \, \frac{t_{21}^2}{2} \Big\} q_{i_1}(t_1) \ddot{q}_{i_2}(t_1 - \epsilon) \Big],
\end{aligned} \tag{127}
$$

$$
\begin{aligned}
W_{GSK}^{(4)} \approx -i \sum_{i_1,\cdots,i_4=1}^{4} \int_{t_0}^{t_f} dt_1 \Big[ \Big\{ \int_{t_0}^{t_1} dt_2 \int_{t_0}^{t_2} dt_3 \int_{t_0}^{t_3} dt_4 \, \langle \mathcal{T}_C O_{i_1}(t_1) O_{i_2}(t_2) O_{i_3}(t_3) O_{i_4}(t_4) \rangle_c \Big\} \\
q_{i_1}(t_1) q_{i_2}(t_1 - \epsilon) q_{i_3}(t_1 - 2\epsilon) q_{i_4}(t_1 - 3\epsilon) \\
+ \Big\{ \int_{t_0}^{t_1} dt_2 \int_{t_0}^{t_2} dt_3 \int_{t_0}^{t_3} dt_4 \, \langle \mathcal{T}_C O_{i_1}(t_1) O_{i_2}(t_2) O_{i_3}(t_3) O_{i_4}(t_4) \rangle_c \, t_{21} \Big\} \\
q_{i_1}(t_1) \dot{q}_{i_2}(t_1 - \epsilon) q_{i_3}(t_1 - 2\epsilon) q_{i_4}(t_1 - 3\epsilon) \\
+ \Big\{ \int_{t_0}^{t_1} dt_2 \int_{t_0}^{t_2} dt_3 \int_{t_0}^{t_3} dt_4 \, \langle \mathcal{T}_C O_{i_1}(t_1) O_{i_2}(t_2) O_{i_3}(t_3) O_{i_4}(t_4) \rangle_c \, t_{31} \Big\} \\
q_{i_1}(t_1) q_{i_2}(t_1 - \epsilon) \dot{q}_{i_3}(t_1 - 2\epsilon) q_{i_4}(t_1 - 3\epsilon) \\
+ \Big\{ \int_{t_0}^{t_1} dt_2 \int_{t_0}^{t_2} dt_3 \int_{t_0}^{t_3} dt_4 \, \langle \mathcal{T}_C O_{i_1}(t_1) O_{i_2}(t_2) O_{i_3}(t_3) O_{i_4}(t_4) \rangle_c \, t_{41} \Big\} \\
q_{i_1}(t_1) q_{i_2}(t_1 - \epsilon) q_{i_3}(t_1 - 2\epsilon) \dot{q}_{i_4}(t_1 - 3\epsilon) \Big].
\end{aligned} \tag{128}
$$

One can compare the leading order forms of the 4-point OTO cumulants of the particle obtained from this approximately local generalised influence phase, and the similar forms obtained from the out of time ordered 1-PI effective action given in section 4.2. This comparison yields the leading order form of any quartic OTO coupling $g$ in terms of an integral of the 4-point OTO cumulants of $O(t)$ as follows

$$
g = \lambda^4 \lim_{t_1 - t_0 \to \infty} \int_{t_0}^{t_1} dt_2 \int_{t_0}^{t_2} dt_3 \int_{t_0}^{t_3} dt_4 \, \mathcal{I}[g] + \mathrm{O}(\lambda^6), \tag{129}
$$

where the dependence of $\mathcal{I}[g]$ on cumulants of the bath are given in table 6 for all the OTO couplings. In expressing these cumulants , we follow the conventions introduced in section 3.4. In addition, we represent the cumulants corresponding to double (anti-)commutators of

$O(t)$ as follows:

$$\langle [12][34]\rangle \equiv \langle [O(t_1), O(t_2)][O(t_3), O(t_4)]\rangle_c,$$
$$\langle [12_+][34]\rangle \equiv \langle \{O(t_1), O(t_2)\}[O(t_3), O(t_4)]\rangle_c,$$
$$\langle [12][34_+]\rangle \equiv \langle [O(t_1), O(t_2)]\{O(t_3), O(t_4)\}\rangle_c,$$
$$\langle [12_+][34_+]\rangle \equiv \langle \{O(t_1), O(t_2)\}\{O(t_3), O(t_4)\}\rangle_c, \text{ etc.}$$

(130)

Table 6: Relations between the OTO couplings and the 4-point OTO cumulants of $O(t)$

| $g$ | $\mathcal{I}[g]$ |
|---|---|
| $\kappa_4$ | $\frac{i}{4}\Big[ 2\big(\langle[1234]\rangle - \langle[4321]\rangle\big) - \big(\langle[2314]\rangle - \langle[3241]\rangle + \langle[1423]\rangle - \langle[4132]\rangle\big)\Big]$ |
| $\widetilde{\kappa}_4$ | $6i(\langle[1234]\rangle + \langle[4321]\rangle)$ |
| $\varrho_4$ | $\frac{1}{2}\Big[ -\langle[12_+3_+4_+]\rangle - \langle[43_+2_+1_+]\rangle + \langle[14_+2_+3_+]\rangle + \langle[41_+3_+2_+]\rangle + \langle[2314_+]\rangle + \langle[3241_+]\rangle\Big]$ |
| $\widetilde{\varrho}_4$ | $\frac{1}{2}\Big[ \big(\langle[43_+21]\rangle + \langle[432_+1]\rangle + \langle[4321_+]\rangle\big) - \big(\langle[12_+34]\rangle + \langle[123_+4]\rangle + \langle[1234_+]\rangle\big)\Big]$ |
| $\kappa_{4\gamma}^I$ | $\frac{i}{4}\Big[ (-t_{12}+t_{13}+t_{14})\big(\langle[34_+][12_+]\rangle - \langle[12_+][34_+]\rangle\big)$ $+(t_{12}-t_{13}+t_{14})\big(\langle[24_+][13_+]\rangle - \langle[13_+][24_+]\rangle\big)$ $+(t_{12}+t_{13}-t_{14})\big(\langle[23_+][14_+]\rangle - \langle[14_+][23_+]\rangle\big)\Big]$ |
| $\kappa_{4\gamma}^{II}$ | $\frac{i}{4}\Big[ (-t_{12}+t_{13}+t_{14})\big(\langle[34][12]\rangle - \langle[12][34]\rangle\big)$ $+(t_{12}-t_{13}+t_{14})\big(\langle[24][13]\rangle - \langle[13][24]\rangle\big)$ $+(t_{12}+t_{13}-t_{14})\big(\langle[23][14]\rangle - \langle[14][23]\rangle\big)\Big]$ |
| $\kappa_{4\gamma}^{III}$ | $-\frac{i}{4}\Big[ (-t_{12}+t_{13}+t_{14})\big(\langle[4321]\rangle + \langle[2314]\rangle + \langle[2413]\rangle\big)$ $+(t_{12}-t_{13}+t_{14})\big(\langle[4231]\rangle + \langle[2341]\rangle + \langle[3412]\rangle\big)\Big]$ |
| $\widetilde{\kappa}_{4\gamma}^I$ | $-\frac{i}{2}\Big[ (t_{12}+t_{13}+t_{14})\langle[1234]\rangle + (-t_{12}-t_{13}+3t_{14})\langle[4321]\rangle\Big]$ |
| $\widetilde{\kappa}_{4\gamma}^{II}$ | $\frac{i}{2}\Big[ (t_{12}-t_{13}+t_{14})\big(\langle[14][23]\rangle - \langle[32][41]\rangle\big)$ $+(t_{12}+t_{13}-t_{14})\big(\langle[12][34]\rangle - \langle[43][21]\rangle + \langle[13][24]\rangle - \langle[42][31]\rangle\big)\Big]$ |
| $\varrho_{4\gamma}^I$ | $\Big[ (t_{12}-t_{13}+t_{14})\big(\langle[12][34]\rangle + \langle[43][21]\rangle\big)$ $+(-t_{12}+t_{13}+t_{14})\big(\langle[13][24]\rangle + \langle[42][31]\rangle + \langle[14][23]\rangle - \langle[32][41]\rangle\big)\Big]$ |
| $\varrho_{4\gamma}^{II}$ | $(-t_{12}+t_{13}+t_{14})\big(\langle[12_+][34]\rangle + \langle[43_+][21]\rangle + \langle[12][34_+]\rangle + \langle[43][21_+]\rangle\big)$ $+(t_{12}-t_{13}+t_{14})\big(\langle[13_+][24]\rangle + \langle[42_+][31]\rangle + \langle[13][24_+]\rangle - \langle[42][31_+]\rangle\big)$ $+(t_{12}+t_{13}-t_{14})\big(\langle[14_+][23]\rangle + \langle[32_+][41]\rangle + \langle[14][23_+]\rangle - \langle[32][41_+]\rangle\big)$ |
| $\widetilde{\varrho}_{4\gamma}^I$ | $\frac{1}{2}\Big[ (-t_{12}+t_{13}+t_{14})\big(\langle[12_+34]\rangle - \langle[43_+21]\rangle\big)$ $+(t_{12}-t_{13}+t_{14})\big(\langle[123_+4]\rangle - \langle[432_+1]\rangle\big)$ $+(t_{12}+t_{13}-t_{14})\big(\langle[1234_+]\rangle + \langle[4321_+]\rangle\big)\Big]$ |
| $\widetilde{\varrho}_{4\gamma}^{II}$ | $\frac{1}{2}\Big[ t_{12}\big(\langle[12][34_+]\rangle + \langle[43_+][21]\rangle\big) + t_{43}\big(\langle[43][21_+]\rangle + \langle[12_+][34]\rangle\big)$ $+t_{13}\big(\langle[13][24_+]\rangle + \langle[42_+][31]\rangle\big) + t_{42}\big(\langle[42][31_+]\rangle + \langle[13_+][24]\rangle\big)$ $+t_{14}\big(\langle[14][23_+]\rangle + \langle[32_+][41]\rangle\big) + t_{32}\big(\langle[32][41_+]\rangle + \langle[14_+][23]\rangle\big)\Big]$ |

# D Quartic couplings in the high temperature limit

In this appendix, we provide the expressions of all the quartic effective couplings at the high temperature limit in terms of the 6 spectral functions given in (105). These expressions are obtained from the forms given in tables 2 and 3 by imposing the KMS relations between the spectral functions. Such KMS relations between the bath's correlators were studied in [10] to express the Fourier transforms of all contour-ordered correlators in terms of a different basis of spectral functions which is given below:

$$\rho[1234], \rho[4321], \rho[2314], \rho[12][34], \rho[13][24], \rho[14][23]. \tag{131}$$

Table 7: SK couplings upto leading order in $\beta$

| $g$ | $\widetilde{\mathcal{I}}[g]$ |
|---|---|
| $\overline{\lambda}_4$ | $-\frac{6}{\omega_1\omega_4(\omega_3+\omega_4)}\rho[1234]$ |
| $\zeta_N^{(4)}$ | $\frac{24i}{\beta^3\omega_1^2\omega_2\omega_3\omega_4^2(\omega_1+\omega_3)(\omega_1+\omega_4)(\omega_3+\omega_4)^2}\Big[\omega_1(\omega_1+\omega_4)(\omega_4^2-\omega_2\omega_3)\rho[1234]$ $+\omega_4(\omega_1+\omega_3)(\omega_1+\omega_4)\rho[4321]$ $-\omega_1\omega_4(\omega_1+\omega_3)(\omega_1+\omega_4)\rho[2143]+\omega_1\omega_2(\omega_3^2-\omega_4^2)\rho[2314]$ $-\omega_2\omega_4(\omega_1+\omega_3)\rho[3241]+\omega_1\omega_4(\omega_1+\omega_4)(\omega_3+\omega_4)\rho[3142]\Big]$ |
| $\overline{\zeta}_3$ | $\frac{2i}{\beta\omega_1^2\omega_2\omega_3\omega_4^2(\omega_1+\omega_3)(\omega_1+\omega_4)(\omega_3+\omega_4)^2}\Big[-\omega_1\omega_2\omega_3(\omega_1+\omega_4)\big((\omega_3+\omega_4)(\omega_1+\omega_3)+\omega_4(\omega_1-\omega_4)\big)\rho[1234]$ $+\omega_1\omega_2\omega_4^2(\omega_1+\omega_3)(\omega_1+\omega_4)\rho[2143]$ $-\omega_1\omega_2\omega_3\omega_4(\omega_3^2-\omega_4^2)\rho[2314]+\omega_2\omega_3\omega_4^2(\omega_1+\omega_3)(\omega_3+\omega_4)\rho[3241]$ $-\omega_1\omega_3\omega_4^2(\omega_1+\omega_4)(\omega_3+\omega_4)\rho[3142]\Big]$ |
| $\zeta_\mu^{(2)}$ | $\frac{1}{\beta^2\omega_1^2\omega_2\omega_3\omega_4^2(\omega_1+\omega_3)(\omega_1+\omega_4)(\omega_3+\omega_4)^2}\Big[\omega_1(\omega_1+\omega_4)\big(\omega_3(\omega_4-\omega_1)(\omega_1+\omega_3)+\omega_4(\omega_3+\omega_4)^2\big)\rho[1234]$ $+\omega_4^2(\omega_1+\omega_3)(\omega_1+\omega_4)(\omega_3+\omega_4)\rho[4321]$ $+\omega_1\omega_4(\omega_1+\omega_4)(\omega_1+\omega_3)^2\rho[2143]+\omega_1\omega_2(\omega_3+\omega_4)(\omega_3^2-\omega_4^2)\rho[2314]$ $-\omega_2\omega_4(\omega_1+\omega_3)(\omega_3+\omega_4)^2\rho[3241]+\omega_1\omega_4(\omega_1+\omega_4)(\omega_3+\omega_4)^2\rho[3142]\Big]$ |
| $\overline{\lambda}_{4\gamma}$ | $\frac{2i}{\omega_1^2\omega_4^2(\omega_3+\omega_4)^2}\Big[3(\omega_4-\omega_1)(\omega_3+\omega_4)+2\omega_1\omega_3\Big]\rho[1234]$ |
| $\overline{\zeta}_{3\gamma}$ | $\frac{1}{\beta\omega_1^3\omega_2\omega_3\omega_4^3(\omega_1+\omega_3)(\omega_1+\omega_4)(\omega_3+\omega_4)^2}\Big[-\omega_1\omega_2\omega_3(\omega_1+\omega_4)\big(\omega_2\omega_3(\omega_2+\omega_3)+5\omega_1\omega_4^2-\omega_4^3\big)\rho[1234]$ $-\omega_1\omega_2\omega_4^2(\omega_1+\omega_3)(\omega_1^2-\omega_4^2)\rho[2143]$ $+\omega_1\omega_2\omega_3\omega_4(\omega_3-\omega_4)\big((\omega_1-\omega_4)(\omega_3+\omega_4)+2\omega_1\omega_4\big)\rho[2314]$ $-\omega_2\omega_3\omega_4^2(\omega_1+\omega_3)\big((\omega_1-\omega_4)(\omega_3+\omega_4)+2\omega_1\omega_4\big)\rho[3241]$ $+\omega_1\omega_3\omega_4^2(\omega_1+\omega_4)\big((\omega_1-\omega_4)(\omega_3+\omega_4)+2\omega_1\omega_4\big)\rho[3142]\Big]$ |
| $\zeta_\gamma^{(2)}$ | $\frac{i}{3\beta^2\omega_1^3\omega_2\omega_3\omega_4^3(\omega_1+\omega_3)(\omega_1+\omega_4)(\omega_3+\omega_4)^3}\Big[-\omega_1(\omega_1+\omega_4)\big(\omega_1^3\omega_3(\omega_3+3\omega_4)+\omega_1^2\omega_3(\omega_3^2+11\omega_3\omega_4+10\omega_4^2)$ $-\omega_4^2(2\omega_3^3+4\omega_3^2\omega_4+3\omega_3\omega_4^2+\omega_4^3)+\omega_1\omega_4(7\omega_3^3+12\omega_3^2\omega_4+8\omega_3\omega_4^2+5\omega_4^3)\big)\rho[1234]$ $-\omega_4^2(\omega_1+\omega_3)(\omega_1+\omega_4)(\omega_3+\omega_4)\big(-\omega_4(\omega_3+\omega_4)+\omega_1(3\omega_3+5\omega_4)\big)\rho[4321]$ $+\omega_1\omega_4(\omega_1+\omega_3)(\omega_1+\omega_4)\big(\omega_3\omega_4(\omega_3+\omega_4)+\omega_1^2(\omega_3+3\omega_4)+\omega_1(\omega_3^2+8\omega_3\omega_4+9\omega_4^2)\big)\rho[2143]$ $+\omega_1\omega_2(\omega_3-\omega_4)(\omega_3+\omega_4)^2\big(\omega_1(\omega_3-5\omega_4)+\omega_4(\omega_3+\omega_4)\big)\rho[2314]$ $-\omega_2\omega_4(\omega_1+\omega_3)(\omega_3+\omega_4)^2\big(\omega_1(\omega_3-5\omega_4)+\omega_4(\omega_3+\omega_4)\big)\rho[3241]$ $+\omega_1\omega_4(\omega_1+\omega_4)(\omega_3+\omega_4)^2\big(\omega_1(\omega_3-5\omega_4)+\omega_4(\omega_3+\omega_4)\big)\rho[3142]\Big]$ |

Notice that the spectral functions corresponding to the nested commutators in this basis are identical to three of the spectral functions given in (105). The remaining three spectral functions corresponding to double commutators can be expressed in terms of the spectral functions in our basis as follows:

$$\rho[12][34] = \frac{\big(1+\mathfrak{f}(\omega_1)\big)\big(1+\mathfrak{f}(\omega_2)\big)}{\big(1+\mathfrak{f}(\omega_1)+\mathfrak{f}(\omega_2)\big)}\Big(\rho[1234]+\rho[2143]\Big),$$

$$\rho[13][24] = \frac{\big(1+\mathfrak{f}(\omega_1)\big)\big(1+\mathfrak{f}(\omega_3)\big)}{\big(1+\mathfrak{f}(\omega_1)+\mathfrak{f}(\omega_3)\big)}\Big(\rho[1234]+\rho[2314]+\rho[3142]\Big), \qquad (132)$$

$$\rho[14][23] = \frac{\big(1+\mathfrak{f}(\omega_1)\big)\big(1+\mathfrak{f}(\omega_4)\big)}{\big(1+\mathfrak{f}(\omega_1)+\mathfrak{f}(\omega_4)\big)}\Big(\rho[2314]+\rho[3241]\Big),$$

where $\mathfrak{f}(\omega)$ is the Bose-Einstein distribution function given by

$$\mathfrak{f}(\omega) \equiv \frac{1}{e^{\beta\omega}-1}. \qquad (133)$$

This demonstrates that the spectral functions given in (105) indeed form a basis. We choose to work with this basis rather than the one given in (131) because, in the high temperature

Table 8: OTO couplings upto leading order in $\beta$

| $g$ | $\widetilde{\mathcal{I}}[g]$ |
|---|---|
| $\kappa_4$ | $-\frac{1}{2\omega_1\omega_4(\omega_3+\omega_4)}\Big[2\rho[1234]+\rho[2143]+\rho[3142]+\rho[3241]\Big]$ |
| $\widetilde{\kappa}_4$ | $-\frac{6}{\omega_1\omega_4(\omega_3+\omega_4)}\Big[\rho[1234]+\rho[4321]\Big]$ |
| $\varrho_4$ | $\frac{i}{\beta\omega_1^2\omega_2\omega_3\omega_4^2(\omega_3+\omega_4)}\Big[-\omega_1\big(\omega_4^2+(\omega_1+\omega_3)(\omega_3+\omega_4)\big)\rho[1234]$ $-\omega_4\big(\omega_1^2+(\omega_1+\omega_3)(\omega_3+\omega_4)\big)\rho[4321]$ $+\omega_1\omega_4(\omega_3-\omega_2)\big(\rho[2143]-\rho[3142]\big)-\omega_1\omega_2(\omega_3-\omega_4)\rho[2314]+\omega_4\omega_2(\omega_1-\omega_3)\rho[3241]\Big]$ |
| $\widetilde{\varrho}_4$ | $-\frac{i}{\beta\omega_1^2\omega_2\omega_3\omega_4^2(\omega_3+\omega_4)}\Big[-\omega_1(\omega_4^2+\omega_1\omega_4+\omega_2\omega_3)\rho[1234]-\omega_4(\omega_1^2+\omega_1\omega_4-\omega_2\omega_3)\rho[4321]$ $-\omega_1\omega_4(\omega_1+\omega_4)\big(\rho[2143]+\rho[3142]\big)-\omega_1\omega_2(\omega_3-\omega_4)\rho[2314]+\omega_4\omega_2(\omega_1+\omega_3)\rho[3241]\Big]$ |
| $\kappa_{4\gamma}^I$ | $\frac{6i}{\beta^2\omega_1^2\omega_2\omega_3\omega_4^2(\omega_1+\omega_3)(\omega_1+\omega_4)(\omega_3+\omega_4)^2}\Big[\omega_1(\omega_1+\omega_4)(\omega_4^2-\omega_2\omega_3)\rho[1234]$ $+\omega_4(\omega_1+\omega_3)(\omega_1+\omega_4)(\omega_3+\omega_4)\rho[4321]$ $-\omega_1\omega_4(\omega_1+\omega_3)(\omega_1+\omega_4)\rho[2143]+\omega_1\omega_2(\omega_3^2-\omega_4^2)\rho[2314]$ $+\omega_1\omega_4(\omega_1+\omega_4)(\omega_3+\omega_4)\rho[3142]-\omega_4\omega_2(\omega_1+\omega_3)(\omega_3+\omega_4)\rho[3241]\Big]$ |
| $\kappa_{4\gamma}^{II}$ | $\frac{i}{4\omega_1^2\omega_4^2(\omega_3+\omega_4)^2}\Big[(\omega_1-\omega_4)(\omega_3+\omega_4)\big(\rho[1234]+\rho[2314]+\rho[3142]\big)$ $+(\omega_1+\omega_4)(\omega_3+\omega_4)\big(\rho[2314]+\rho[3241]\big)$ $+\big((\omega_1-\omega_4)(\omega_3+\omega_4)+2\omega_1\omega_4\big)\big(\rho[1234]+\rho[2143]\big)\Big]$ |
| $\kappa_{4\gamma}^{III}$ | $-\frac{i}{4\omega_1^2\omega_4^2(\omega_3+\omega_4)^2}\Big[(\omega_1-\omega_4)(\omega_3+\omega_4)\big(\rho[1234]+\rho[2143]\big)$ $+\big((\omega_1-\omega_4)(\omega_3+\omega_4)+2\omega_1\omega_4\big)\big(\rho[1234]+\rho[3241]+\rho[3142]\big)\Big]$ |
| $\widetilde{\kappa}_{4\gamma}^{I}$ | $\frac{i}{2\omega_1^2\omega_4^2(\omega_3+\omega_4)^2}\Big[\big(-\omega_1(\omega_2+\omega_1)+\omega_4(3\omega_2+5\omega_1)\big)\rho[1234]$ $+\big(-\omega_4(\omega_3+\omega_4)+\omega_1(3\omega_3+5\omega_4)\big)\rho[4321]\Big]$ |
| $\widetilde{\kappa}_{4\gamma}^{II}$ | $\frac{i}{2\omega_1^2\omega_4^2(\omega_3+\omega_4)^2}\Big[(\omega_1-\omega_4)\big(\rho[2314]+\rho[3241]\big)+(\omega_1+\omega_4)\big(2\rho[1234]+\rho[2314]+\rho[2143]+\rho[3142]\big)\Big]$ |
| $\varrho_{4\gamma}^I$ | $\frac{2}{\beta\omega_1^2\omega_4^2(\omega_1+\omega_3)(\omega_1+\omega_4)(\omega_3+\omega_4)^2}\Big[(\omega_1+\omega_4)(\omega_1^2-4\omega_1\omega_4+\omega_4^2)\rho[1234]+(\omega_1+\omega_3)(\omega_1^2-\omega_4^2)\rho[2143]$ $-(\omega_3-\omega_4)\big(\omega_4(\omega_3+\omega_4)-\omega_1(\omega_3+3\omega_4)\big)\rho[2314]$ $-(\omega_1+\omega_3)\big(\omega_4(\omega_3+\omega_4)-\omega_1(\omega_3+3\omega_4)\big)\rho[3241]$ $+(\omega_1+\omega_4)\big(\omega_4(\omega_3+\omega_4)-\omega_1(\omega_3+3\omega_4)\big)\rho[3142]\Big]$ |
| $\varrho_{4\gamma}^{II}$ | $\frac{1}{\beta\omega_1^3\omega_2\omega_3\omega_4^3(\omega_3+\omega_4)^2}\Big[\omega_1\big(-\omega_3\omega_4(\omega_3+\omega_4)^2+\omega_1^2(\omega_3^2+3\omega_3\omega_4+\omega_4^2)+\omega_1(\omega_3^3+2\omega_3^2\omega_4+\omega_3\omega_4^2+\omega_4^3)\big)\rho[1234]$ $+\omega_4(\omega_3+\omega_4)\big(\omega_1^3+(\omega_1^2+\omega_1\omega_3-\omega_3\omega_4)(\omega_3+\omega_4)\big)\rho[4321]$ $-\omega_1\omega_4\big(\omega_1^2\omega_4-\omega_4(\omega_3+\omega_4)^2+\omega_1\omega_3(\omega_3+2\omega_4)\big)\rho[2143]$ $-\omega_1^2\omega_2(\omega_3+\omega_4)^2\rho[2314]+\omega_2\omega_4(\omega_3+\omega_4)(-\omega_1^2+\omega_3\omega_4)\rho[3241]$ $+\omega_1\omega_4(\omega_3+\omega_4)\big(\omega_1^2-\omega_3\omega_4+\omega_1(\omega_3+\omega_4)\big)\rho[3142]\Big]$ |
| $\widetilde{\varrho}_{4\gamma}^{I}$ | $\frac{1}{\beta\omega_1^3\omega_2\omega_3\omega_4^3(\omega_3+\omega_4)^2}\Big[\omega_1\big(\omega_1^2(\omega_3^2+2\omega_3\omega_4+3\omega_4^2)+\omega_4(\omega_3^3+2\omega_3^2\omega_4-\omega_4^3)+\omega_1(\omega_3^3+3\omega_3^2\omega_4+4\omega_3\omega_4^2+2\omega_4^3)\big)\rho[1234]$ $+\omega_4\big(\omega_3\omega_4(\omega_3+\omega_4)^2+\omega_1^3(\omega_3+3\omega_4)+\omega_1^2(\omega_3^2+3\omega_3\omega_4+2\omega_4^2)+\omega_1(\omega_3^3+3\omega_3^2\omega_4+\omega_3\omega_4^2-\omega_4^3)\big)\rho[4321]$ $+\omega_1\omega_4(\omega_3+\omega_4)(\omega_1^2-\omega_4^2)\rho[2143]-\omega_1\omega_2(\omega_3-\omega_4)\big(\omega_4(\omega_3+\omega_4)-\omega_1(\omega_3+3\omega_4)\big)\rho[2314]$ $+\omega_2\omega_4\big(\omega_3\omega_4(\omega_3+\omega_4)-\omega_1^2(\omega_3+3\omega_4)+\omega_1(-\omega_3^2-2\omega_3\omega_4+\omega_4^2)\big)\rho[3241]$ $+\omega_1\omega_4(\omega_1+\omega_4)\big(-\omega_4(\omega_3+\omega_4)+\omega_1(\omega_3+3\omega_4)\big)\rho[3142]\Big]$ |
| $\widetilde{\varrho}_{4\gamma}^{II}$ | $\frac{1}{\beta\omega_1^2\omega_2\omega_3\omega_4^2(\omega_3+\omega_4)^2}\Big[\big(2\omega_3^3+4\omega_3^2\omega_4+\omega_3\omega_4^2-\omega_4^3+\omega_1^2(\omega_3+4\omega_4)+\omega_1(\omega_3^2+3\omega_3\omega_4+3\omega_4^2)\big)\rho[1234]$ $+\big(2\omega_3^3+5\omega_3^2\omega_4+2\omega_3\omega_4^2-\omega_4^3+2\omega_1^2(\omega_3+2\omega_4)+\omega_1(2\omega_3^2+5\omega_3\omega_4+3\omega_4^2)\big)\rho[4321]$ $+(\omega_1-\omega_3-\omega_4)(\omega_1+\omega_4)(\omega_3+2\omega_4)\rho[2143]+\omega_2(-\omega_3^2-4\omega_1\omega_4+\omega_4^2)\rho[2314]$ $+\omega_2\big(-\omega_3^2-2\omega_3\omega_4+\omega_4^2-2\omega_1\omega_3-4\omega_1\omega_4\big)\rho[3241]$ $+(\omega_1+\omega_4)\big(\omega_3^2-\omega_4^2+2\omega_1\omega_3+4\omega_1\omega_4\big)\rho[3142]\Big]$ |

limit, the $\beta$-expansions of all the spectral functions in this basis begin at the same power of $\beta$. This simplifies comparisons between the leading order forms of the different couplings.

Now, let us provide the expressions of the quartic couplings in terms of this basis. In the high temperature limit, any quartic coupling can be expressed as

$$g = \int_{\mathcal{C}_4} \widetilde{\mathcal{I}}[g] + O(\lambda^6), \tag{134}$$

where the integrand $\widetilde{\mathcal{I}}[g]$ for the Schwinger-Keldysh couplings and the OTO couplings are

given in tables 7 and 8 respectively. In these integrands, the spectral functions are truncated at their leading order in $\beta$-expansion.

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
