# Peer review of "Out of time ordered effective dynamics of a quartic oscillator"

_SciPost Physics, doi:SciPost Phys. 7, 013 (2019)_

## Round 3 · Referee Report · Anonymous (Referee 1) · 2019-7-3

Strengths

1 - This paper is very well written.
2 - It provides a useful introduction and overview of recent progress in formulating consistent effective actions for a Brownian particle which incorporate out of time ordered correlators and is developed up to a quartic order in the field expansion.
3 - It derives genuinely new generalized Onsager and fluctuation-dissipation relations among certain coefficients of the quartic Schwinger-Keldysh couplings.
4 - The authors also illustrate the findings with an instructive explicit example.

Weaknesses

  1. The new generalized Onsager and fluctuation-dissipation relations are among quartic couplings, thus, at this stage, it is not clear how widespread of a role they might play in applications and how one can (experimentally) verify them.

Report

The paper addresses a difficult question of classifying all possible couplings arising at the quartic order in the field expansion of the Schwinger-Keldysh effective action for a Brownian particle and its out of time ordered generalization.

By carefully analyzing the constraints imposed on this effective action by the time reversibility and thermality of the bath to which the quantum Brownian particle is coupled, the authors are able to identify novel Onsager and fluctuation-dissipation relations on certain quartic couplings.

This result is sufficiently interesting for the paper to be published. Moreover, given that the paper is written very clearly, is well structured and nicely formatted, I can recommend for the paper to be published without any changes apart from some small typos and one question enumerated below.

Requested changes

Questions:

1 - Have you not included the OTO couplings of the generalized Schwinger-Keldysh effective action, would you have found the generalized fluctuation-dissipation relations among the quartic Schwinger-Keldysh couplings?

Typos:

1 - Introduction: last sentence of the first paragraph:
"... for only the degrees of freedom of the system."
Perhaps: "... for only the INFRARED (LOW-ENERGY) degrees of freedom of the system." ?
2 - Below (19):
"Plugging these expression..."
perhaps "Plugging these expressions..." ?
3 - Caption of Figure 5: "t?0" perhaps $t>0$?

---

## Round 5 · Author Response

Dear Editor,

We thank the referee for his/her comments.

We will begin by addressing the referee's question on whether we could have obtained the generalised fluctuation-dissipation relation between the quartic Schwinger-Keldysh couplings without including the OTO couplings in the analysis. Our response is as follows:

The generalised fluctuation-dissipation relation essentially relies on certain relations between the bath's correlators due to a combination of microscopic reversibility and thermality. Therefore, in principle, one could use these relations between the bath's correlators to derive the generalised fluctuation-dissipation relation without introducing the OTO couplings. However, as we have argued in our paper, studying these relations between the correlators of the bath requires one to include the bath's OTOCs in the analysis. As the OTO couplings of the particle encode the effects of the bath's OTOCs on the particle's dynamics, introducing them simplifies the analysis. As we have shown, these OTO couplings allow one to study the constraints imposed by the bath's microscopic reversibility and thermality separately. Combining these constraints, one can easily get the generalised fluctuation-dissipation relation.

Now, we would like to mention that we have fixed the typos pointed out by the referee in his/her report. In addition, we noticed two more minor errors in the previous version which we enumerate below:

1) In equations (5), (7) and (9), we made a mistake while writing the quartic combinations of the couplings between the particle and the bath oscillators which contribute to the cumulants of the 4-point correlators of O (the bath operator that couples to the particle). We have corrected them in the current version. One can verify that these corrected combinations indeed contribute to the cumulants of the 4-point correlators of O by looking at the Feynman diagrams given in Figure 9 of appendix A.2.

2) We also made a mistake while showing the positions of the insertions in figure 3. We have corrected this in the current version. It is now consistent with the correlator mentioned in the caption of the same figure.

Sincerely, the Authors

---

## Round 5 · List of Changes

1) In the last line of the first paragraph of the introduction, we have replaced the phrase "degrees of freedom of the system" by "infrared (low-frequency) degrees of freedom of the system".

2) In the line just below equation (19) in page 10, we have replace the phrase "Plugging these expression" by "Plugging these expressions".

3) In the the caption of figure 5, we have replaced "t?0" by "t>0".

4) In equations (5), (7) and (9), we have corrected the combination of the quartic couplings between the particle and the bath oscillators which contribute to the cumulants of the 4-point correlators of O (the bath operator that couples to the particle).

5) In figure 3, we have corrected the positions of the insertions.

6) We have updated the references.

---

## Editorial Decision

published